# Multiple sources of aerobic methane production in aquatic ecosystems include bacterial photosynthesis

Elisabet Perez-Coronel [1] & J. Michael Beman [1] ✉

Aquatic ecosystems are globally significant sources of the greenhouse gas methane to the atmosphere. Until recently, methane production was thought to be a strictly anaerobic process confined primarily to anoxic sediments. However, supersaturation of methane in oxygenated waters has been consistently observed in lakes and the ocean (termed the 'methane paradox'), indicating that methane can be produced under oxic conditions through unclear mechanisms. Here we show aerobic methane production from multiple sources in freshwater incubation experiments under different treatments and based on biogeochemical, metagenomic, and metatranscriptomic data. We find that aerobic methane production appears to be associated with (bacterio)chlorophyll metabolism and photosynthesis, as well as with Proteobacterial degradation of methylphosphonate. Genes encoding pathways for putative photosynthetic- and methylphosphonate-based methane production also co-occur in Proteobacterial metagenome-assembled genomes. Our findings provide insight into known mechanisms of aerobic methane production, and suggest a potential co-occurring mechanism associated with bacterial photosynthesis in aquatic ecosystems.

Atmospheric concentrations of the potent greenhouse gas methane ($CH_4$) have increased significantly due to anthropogenic activity, representing an important component of climate change[1]. However, these increases are superimposed on substantial spatial and temporal variability in natural sources of $CH_4$ to the atmosphere. Of all natural $CH_4$ sources, freshwater lakes are particularly important but poorly understood, with their estimated contribution ranging from 6 to 16% of all natural $CH_4$ emissions—despite accounting for only ~0.9% of Earth's surface area[2]. $CH_4$ emissions from lakes are conventionally viewed to be regulated by $CH_4$ production occurring predominantly in anoxic sediments, followed by subsequent $CH_4$ oxidation in surface sediments and the water column[3]. However, supersaturation of $CH_4$ has been consistently observed in oxygenated waters of aquatic systems[4]. This observation may indicate that $CH_4$ is produced under oxic conditions, and that the rate of oxic $CH_4$ production exceeds $CH_4$ oxidation. Since archaeal methanogenesis is an obligate anaerobic process[5], the accumulation of $CH_4$ under oxic conditions is typically

referred to as the 'methane paradox,' and has been observed in oceans[6,7], lakes[8-10], and aerobic wetland soils[11]. Notably, aerobic $CH_4$ production occurs near the surface, and so any produced $CH_4$ may readily flux to the atmosphere. Identifying which mechanisms produce $CH_4$ in oxygenated waters is therefore essential for our understanding of $CH_4$ fluxes and their contribution to climate change.

Although multiple mechanisms for paradoxical (i.e., aerobic) $CH_4$ production have been proposed, the degree to which these are active in freshwater lakes remains unknown. Initial studies suggested that $CH_4$ production under oxygenated conditions could occur in anoxic microsites present in the water column—such as fecal pellets, detritus, and the gastrointestinal tracts of larger organisms such as fish or zooplankton[12-14]. Several studies have also demonstrated a correlation between phytoplankton or primary production and $CH_4$ production[8-10]. However, the underlying reason(s) for this relationship is unclear. One possibility is that methanogens reside on the surface of phytoplankton cells and produce $CH_4$ in anoxic microsites[8]. Bogard

[1]Environmental Systems and Sierra Nevada Research Institute, University of California Merced, Merced, CA, USA. ✉e-mail: jmbeman@gmail.com

et al.[9] and Donis et al.[15] also hypothesized that several groups of methanogens have oxygen-tolerant or detoxifying pathways that could aid in $CH_4$ production in the presence of oxygen.

In contrast, the current prevailing view of marine ecosystems is that methylphosphonate (MPn) is the main precursor of $CH_4$ production under oxic conditions–particularly in phosphorus (P)-stressed ecosystems such as the open ocean[6,16]. MPn is the simplest form of organic carbon (C)-P bonded compounds in aquatic ecosystems; microbial utilization of MPn as a source of P, and the consequent breakdown of the C-P bond, releases $CH_4$ as a by-product[6,16–18]. A broad range of marine and freshwater bacteria have the genomic potential to metabolize MPn and produce $CH_4$, based on the presence of the multi-gene C-P lyase pathway in their genomes. This includes multiple groups of Proteobacteria, Firmicutes, Bacteroidetes, Chloroflexi, and Cyanobacteria[17–19]. While expression of this pathway is thought to be regulated by P availability[17–19], how widely this occurs in freshwater ecosystems is not well known[20–22]. Recent work[23] also indicates that

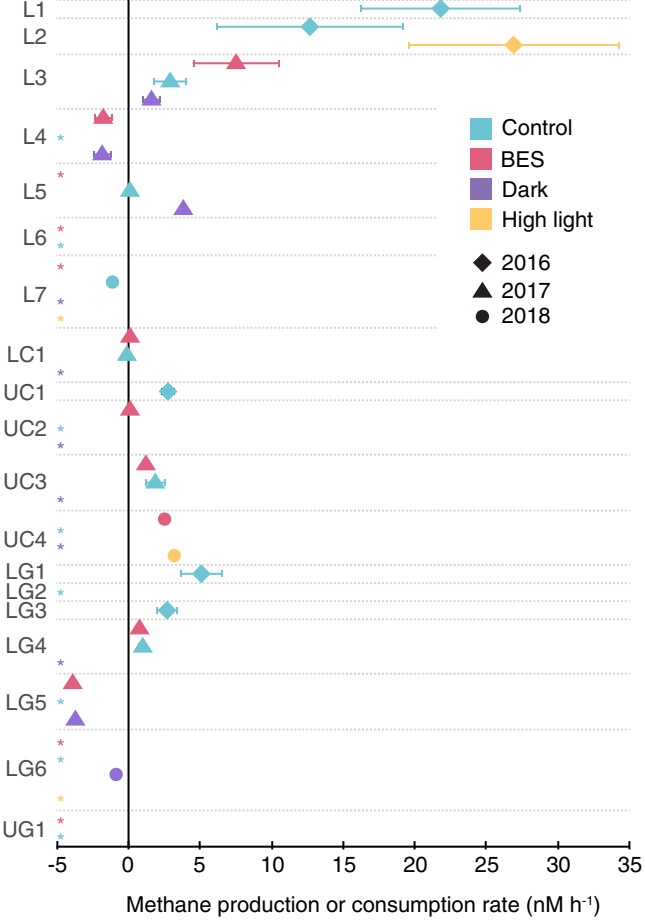

**Fig. 1 | Methane ($CH_4$) production or consumption rates across experiments.** Different colors represent different treatments or control incubations. Different symbols denote the year of the experiment. Abbreviations at left denote lake (Lukens =L, Lower Cathedral =LC, Upper Cathedral =UC, Lower Gaylor =LG, and Upper Gaylor =UG) and experiment number in each lake (e.g., UC4 is the fourth experiment in Upper Cathedral Lake). Rates were calculated from the slopes of significant ($P < 0.05$) correlations between time and triplicate $CH_4$ concentration measurements conducted at multiple timepoints during experimental incubations. Where symbols are not shown, asterisks indicate that no significant production or consumption occurred. Error bars denote standard errors of slope values for significant correlations between $CH_4$ and time; in some cases, these are smaller than the datapoints and are not visible. BES indicates the addition of the methanogenesis inhibitor 2-bromoethanesulphonate. Source data are provided as a Source Data file.

$CH_4$ can be produced aerobically via methylamine (MeA) metabolism in freshwater, and that $CH_4$ production can switch from being associated with MeA to MPn metabolism with a week's time–highlighting the importance of examining the mechanisms underlying paradoxical $CH_4$ production.

Finally, cultures of marine and freshwater Cyanobacteria can directly produce $CH_4$[24]. However, outside of a single experiment[25], this has not been examined in aquatic ecosystems. More significantly, the exact mechanism by which this occurs remains unknown, although it occurs during exposure to light[24]. Given the widespread distribution of cyanobacteria in the ocean and freshwater, identifying the potential mechanism(s) by which cyanobacteria produce $CH_4$–and whether this capability may be present in other photosynthetic organisms–is of broad relevance.

Together these proposed mechanisms for $CH_4$ production–(i) methanogenesis aided by detoxifying genes or in anoxic microsites, (ii) $CH_4$ production by breakdown of methylated compounds, and (iii) $CH_4$ production by Cyanobacteria–point to multiple mechanisms by which $CH_4$ can be produced under oxygenated conditions. Paradoxical $CH_4$ production is also notably variable–ranging over several orders of magnitude within weeks[26]–which may reflect the relative activity of individual $CH_4$ production mechanisms. However, many of these mechanisms are recently discovered and therefore poorly understood, and so the degree to which they occur within different aquatic ecosystems is largely unknown.

Here we use an interdisciplinary approach to disentangle these mechanisms and determine which may produce $CH_4$ in freshwater lakes. In order to rule out physical transport and focus on potential biological mechanisms of oxic $CH_4$ production, we conduct incubation experiments using surface waters from mountain lakes located along an elevation gradient in Yosemite National Park in California (Figure S1, Table S1). We investigate specific mechanisms using a combination of $CH_4$ measurements over time, experimental treatments and inhibitors, and 16S rRNA gene and transcript sequencing, while also applying stable isotope analyses and metagenome and metatranscriptome sequencing to selected experiments. Paradoxical $CH_4$ production is evident in multiple experiments and experimental treatments, as is MPn breakdown via widely-distributed members of the *Comamonadaceae* family. However, experimental treatments, stable isotope $\delta^{13}C$ signatures of $CH_4$, and metatranscriptomic data also point to a new potential mechanism of paradoxical $CH_4$ production carried out by photosynthetic bacteria.

## Results

Our experiments provide multiple lines of evidence for paradoxical $CH_4$ production in freshwater lakes. Out of 19 total experiments conducted in five lakes, 63% showed $CH_4$ production in unamended controls and/or at least one experimental treatment (see below; based on $CH_4$ concentration measurements in replicate bottles incubated for at least 24 h; Fig. 1). We observed the highest $CH_4$ production rates in Lukens Lake (e.g., L1 and L2 experiments), as well as consistent production in Lower Gaylor Lake (LG1, LG3, and LG4) and Upper Cathedral Lake (all UC experiments). Net $CH_4$ production rates ranged from 0.086 to 26.9 nM $h^{-1}$, with the majority of values <4 nM $h^{-1}$. These rates are consistent with the limited experiments previously conducted in other freshwater lakes–e.g., 0.1–10.8 nM $h^{-1}$ in Lake Stechlin, Germany[8,10,26]–and, on the higher end, our values are similar to the range reported by Bogard et al.[9] for experimental manipulations in Lac Cromwell (2.08–8.33 nM $h^{-1}$). $CH_4$ turnover rates also ranged from 5 to 146 days (with in situ concentrations ranging from 309-2839 nM), consistent with turnover rates of ~18 days in Lake Stechlin ($CH_4$ concentration ~430 nM; ref. 8), ~2.2 days in Lac Cromwell ($CH_4$ concentration ~200 nM; ref. 9), and 67 days in Yellowstone Lake ($CH_4$ concentration of 46.3 nM; ref. 21).

Experiments that were ca. 24 h in length frequently showed paradoxical CH$_4$ production (Fig. 1, Table S2). (The only exception was the LG2 experiment, which was not significant due to variation among replicates). We subsequently increased the length and number of experimental treatments with the intention of capturing variations in the balance between co-occurring CH$_4$ production and consumption over time. For example, initial CH$_4$ production could be followed by oxidation once CH$_4$ concentrations reach a certain threshold required for oxidation[8]. Conversely, initial decreases due to oxidation may be followed by eventual production—for example through the development of P limitation that triggers CH$_4$ production via MPn metabolism[6]. We tested for nonlinear patterns using piecewise regression, and found two experiments with initial CH$_4$ production followed by oxidation (LG1 and UC3 control), as well as two exhibiting oxidation followed by production (LC1 and UC2; Supplementary Note 1). It is also possible that production and oxidation proceed at similar rates, leading to no net change in concentrations, even though CH$_4$ is actively cycled. The L6 and UG1 experiments were among the few experiments without detectable changes in CH$_4$ and may reflect this balance between production and oxidation (or a lack of CH$_4$ cycling entirely). In this case, molecular data are useful for providing insight into the underlying dynamics. We consequently examined expression of the particulate methane monooxygenase gene *(pmoA)*, and found that *pmoA* was expressed in the majority of the incubations surveyed (Supplementary Note 1). Finally, we found that the longer, 96-hour L7 and LG6 experiments in 2018 showed net methane oxidation.

Observed decreases in CH$_4$ concentrations in some experiments, as well as omic data, are indicative of active CH$_4$ oxidation that may obscure paradoxical production. Put another way, CH$_4$ is clearly produced in 63% of experiments based on CH$_4$ measurements alone, but may also occur in experiments showing no significant change (or even net consumption) over time. We therefore evaluated potential paradoxical CH$_4$ production mechanisms across experiments using multiple experimental treatments (a methanogenesis inhibitor, dark conditions, and high light intensity) and analyses.

## Methanogenesis

We used these approaches to test for phytoplankton- or particle-based methanogenesis, as initial observations in freshwater lakes showed methanogens attached to phytoplankton[8], while other work has suggested CH$_4$ production can occur in anoxic microsites on particles[12–14]. In all experiments from 2017 and 2018, we included a treatment that consisted of the addition of the methanogenesis inhibitor BES (2-bromoethanesulphonate). While there are some caveats with its use[27], BES is widely used, including earlier tests of the CH$_4$ paradox[8,10]. Despite the addition of a known methanogenesis inhibitor, paradoxical CH$_4$ production rates in four BES treatments (L3, L4, UC3, UC4) were significantly higher than CH$_4$ production in controls (Fig. 1 and S2). Two other BES treatments showed CH$_4$ consumption followed by significant production (LC1 and UC2), while two showed significant decreases in CH$_4$ over time (L4 and LG5). The remaining five BES treatments showed no significant trends (Fig. S2). In parallel, 16S rRNA sequencing of DNA from experiments L1-6 and LG1-4 recovered no methanogen 16S sequences (out of 5.4 million total sequences; Table 1). Sequencing of 16S rRNA transcripts in RNA samples from experiments L5-6, LG5-6, and UC3-4 recovered methanogen sequences in only two samples (dark treatments from L5 and LG5) at very low levels (Table 1). Analysis of metatranscriptomes likewise showed that methyl coenzyme M reductase (*mcrA*; responsible for the final step in methanogenesis) transcripts were absent in LG6 and UC4(tf) incubations, and present at low levels in LG5, L7 and UC4(t0) (Fig. 2a). *mcrA* genes were absent from all metagenomes except the L6(tf) incubation (Fig. 2b). Following earlier work[8,28], 16S rRNA and *mcrA* sequences were affiliated with the *Methanosaeta* and *Methanospirillum* genera,

indicating the potential for hydrogenotrophic and acetoclastic (but not methylotrophic) methanogenesis[28]. (Incubations were also monitored to confirm that they were under oxic conditions at all times; Table S1.) Collectively these data provide limited evidence for water column-based methanogenesis as source of CH$_4$ in the LG5 experiment, as BES reduced CH$_4$ production (Fig. 1), and methanogen 16S rRNA (Table 1) and *mcrA* genes (Fig. 2) were expressed at low levels. However, in many other experiments, CH$_4$ was produced while methanogens were almost entirely absent and inactive based on 16S rRNA and *mcrA* genes and transcripts (Table 1; Fig. 2).

## Aerobic methylphosphonate and methylamine metabolism as sources of methane

Instead, analysis of metatranscriptomes and metagenomes showed potential for multiple additional CH$_4$ production mechanisms. In particular, we found evidence for microbial metabolism of MPn based on the universal expression of alpha-D-ribose 1-methylphosphonate 5-phosphate C-P lyase (*phnJ*) transcripts in metatranscriptomes (Fig. 2a). *phnJ* is responsible for the cleavage of the C-P bond that results in CH$_4$ production from MPn; this mechanism is thought to be the dominant paradoxical CH$_4$ production mechanism in the ocean[16], but has been documented in only a limited number of freshwater lakes[20–22]. Along with *phnJ*, phosphonate-binding periplasmic protein (*phnD*) genes (involved in the binding component of phosphonate uptake[18]) were also expressed in all metatranscriptomes (Fig. 2a). *phnJ* and *phnD* genes were also recovered in all metagenomes (Fig. 2b). Finally, we examined whether genes involved in the production of MPn were also present and expressed. Transcripts of phosphoenolpyruvate mutase (*pepM*) and phosphonopyruvate decarboxylase (*ppd*)—involved in phosphonate biosynthesis[29]—were present in some metatranscriptomes (Fig. 2), suggesting that phosphonates may be synthesized in situ.

Overall, over 40 different bacterial genera expressed or possessed *phnJ* genes, indicating that P assimilation from MPn is broadly distributed among surface water microbes. However, *phnJ* transcripts and genes were most commonly found among organisms in the betaproteobacterial *Comamonadaceae* family, with multiple *Comamonadaceae* genera accounting for 20.9% of *phnJ* transcripts in metatranscriptomes and 27.4% of *phnJ* genes in metagenomes (Table S3). *phnJ* transcripts and genes from *Sphingobacteriales* were abundant in several samples (LG5 and LG6 metatranscriptomes, L6 metagenomes), while *Comamonadaceae phnJ* transcripts and genes were most abundant in every other sample. *Comamonadaceae phnJ* transcripts and genes showed notably high identity to database sequences, with the majority of sequences showing >92% and up to 100% amino acid identity to *Acidovorax, Hydrogenophaga, Limnohabitans, Polaromonas, Rhodoferax,* and *Variovorax phnJ* database sequences (Table S3).

In parallel with paradoxical CH$_4$ production from MPn, recent work in Yellowstone Lake has also implicated *Acidovorax* in CH$_4$ production from MeA via pyridoxal 5′ phosphate-dependent aspartate aminotransferase (*aat*; ref. 23). Additional isolates with this ability included *Pseudomonas, Caulobacter, Mesorhizobium,* and *Dietzia* spp[23].—but this paradoxical CH$_4$ production mechanism has only been examined in a single location. We found that *aat* sequences from four of these five groups (*Acidovorax, Pseudomonas, Caulobacter,* and *Mesorhizobium*) were present in metatranscriptomes and metagenomes from Yosemite (Table S4). While some *aat* sequences were affiliated with *Acidovorax, Caulobacter aat* sequences were more common (Table S4). We also BLASTed the *aat* sequence from the Yellowstone *Acidovorax* isolate against our metatranscriptomes and metagenomes; in line with Wang et al.[23], we found that similar *aat* sequences in Yosemite were affiliated with *Polaromonas* and *Limnohabitans*. Overall, *aat* genes and transcripts from these groups were present in similar proportions to *phnJ*.

**Table 1 | Relative abundance of methanogens, Cyanobacteria, *Comamonadaceae*, and *Caulobacter* as a percentage of 16S rRNA genes and transcripts sequenced in different experiments and treatments (where BES indicates addition of the methanogenesis inhibitor 2-bromoethanesulphonate)**

| Experiment | Treatment | Time (h) | Methanogens | Cyanobacteria | *Comamonadaceae* | *Caulobacter* |
|---|---|---|---|---|---|---|
| | | | **% of 16S rRNA genes** | | | |
| L1 | Control | 0 | 0 | 0.902 | 9.17 | 0.568 |
| L2 | Control | 0 | 0 | 1.10 | 6.03 | 1.01 |
| | Control | 24 | 0 | 1.65 | 5.20 | 0.767 |
| L3 | Control | 0 | 0 | 0.061 | 4.56 | 0.356 |
| L4 | Control | 0 | 0 | 2.38 | 7.97 | 0.640 |
| L6 | Control | 0 | 0 | 5.55 | 2.03 | 0.542 |
| | BES | 45 | 0 | 12.6 | 0.338 | 0.018 |
| LG1 | Control | 0 | 0 | 0.646 | 5.32 | 1.09 |
| | Control | 24 | 0 | 0.377 | 2.04 | 0.452 |
| LG2 | Control | 0 | 0 | 0.795 | 11.1 | 0.530 |
| LG3 | Control | 24 | 0 | 0.246 | 0.881 | 3.20 |
| LG4 | Control | 0 | 0 | 5.79 | 4.35 | 0.263 |
| UC1 | Control | 0 | 0 | 0.880 | 11.6 | 0.156 |
| | | | **% of 16S rRNA transcripts** | | | |
| L5 | Control | 0 | 0 | 46.0 | 6.43 | 0.439 |
| | Control | 74 | 0 | 28.8 | 6.66 | 0.873 |
| | Dark | 74 | 0.014 | 58.5 | 4.70 | 1.11 |
| | BES | 74 | 0 | 24.9 | 8.65 | 0.423 |
| L6 | Control | 0 | 0 | 8.34 | 6.27 | 0 |
| | Control | 55 | 0 | 0.800 | 5.42 | 0 |
| | BES | 55 | 0 | 0 | 5.20 | 1.84 |
| L7 | Dark | 96 | 0 | 65.5 | 0.841 | 0 |
| | BES | 96 | 0 | 46.4 | 1.95 | 0.161 |
| | High light | 96 | 0 | 45.0 | 1.06 | 0 |
| LG5 | Control | 0 | 0 | 17.3 | 12.8 | 0.374 |
| | Control | 86 | 0 | 2.00 | 12.0 | 1.42 |
| | Dark | 86 | 0.008 | 19.2 | 10.2 | 0.411 |
| | BES | 86 | 0 | 1.04 | 25.6 | 2.41 |
| LG6 | Control | 0 | 0 | 3.05 | 6.14 | 0.277 |
| | Control | 96 | 0 | 2.45 | 8.69 | 0.376 |
| | Dark | 96 | 0 | 2.65 | 2.53 | 1.34 |
| | BES | 96 | 0 | 2.28 | 5.02 | 0.498 |
| | High light | 96 | 0 | 7.18 | 2.64 | 2.34 |
| UC3 | Control | 0 | 0 | 0.227 | 7.98 | 1.53 |
| | Control | 24 | 0 | 0.275 | 8.72 | 0.439 |
| | Control | 48 | 0 | 0.221 | 15.2 | 0.793 |
| | Control | 57 | 0 | 0 | 12.3 | 0 |
| | Dark | 57 | 0 | 0 | 8.66 | 1.44 |
| | BES | 57 | 0 | 0 | 10.9 | 2.80 |
| UC4 | Control | 96 | 0 | 3.148 | 14.0 | 1.27 |
| | High light | 96 | 0 | 0.474 | 7.25 | 1.35 |

16S rRNA gene and transcript data were consistent with functional gene data from metatranscriptomes and metagenomes. *Comamonadaceae* (which includes *Acidovorax, Polaromonas*, and *Limnohabitans*) were universally present and abundant across experiments, while *Caulobacter* were consistently present in lower proportions. In 16S rRNA gene sequence libraries from the L1-6, LG1-4, and UC1 experiments, *Comamonadaceae* ranged from 2.04% to 11.6% of all sequences (Table 1), and were abundant in several incubations with significant CH$_4$ production (L1, L2, UC1). Sequencing of 16S rRNA transcripts in RNA samples from experiments L5-6, LG5-6, and UC3-4 showed similar values (Table 1), and *Comamonadaceae* reached 15% of all 16S rRNA transcripts in the UC3 and UC4 experiments (where CH$_4$ was significantly produced in several treatments). Similar to *phnJ* results, 16S rRNA sequences from the dominant *Comamonadaceae* ASVs showed high (98.8-100%) nucleotide sequence identity to 16S rRNA genes in sequenced genomes (Table S5). In comparison with the *Comamonadaceae*, *Caulobacter* ranged from 0% to 3.18% of 16S rRNA genes and transcripts and were typically less abundant in 16S rRNA gene libraries (with the exception of the LG3 sample; Table 1). *Caulobacter* reached >2% of 16S rRNA transcripts in some treatments in the LG5, LG6, and UC3 experiments, with the latter showing significant paradoxical CH$_4$ production.

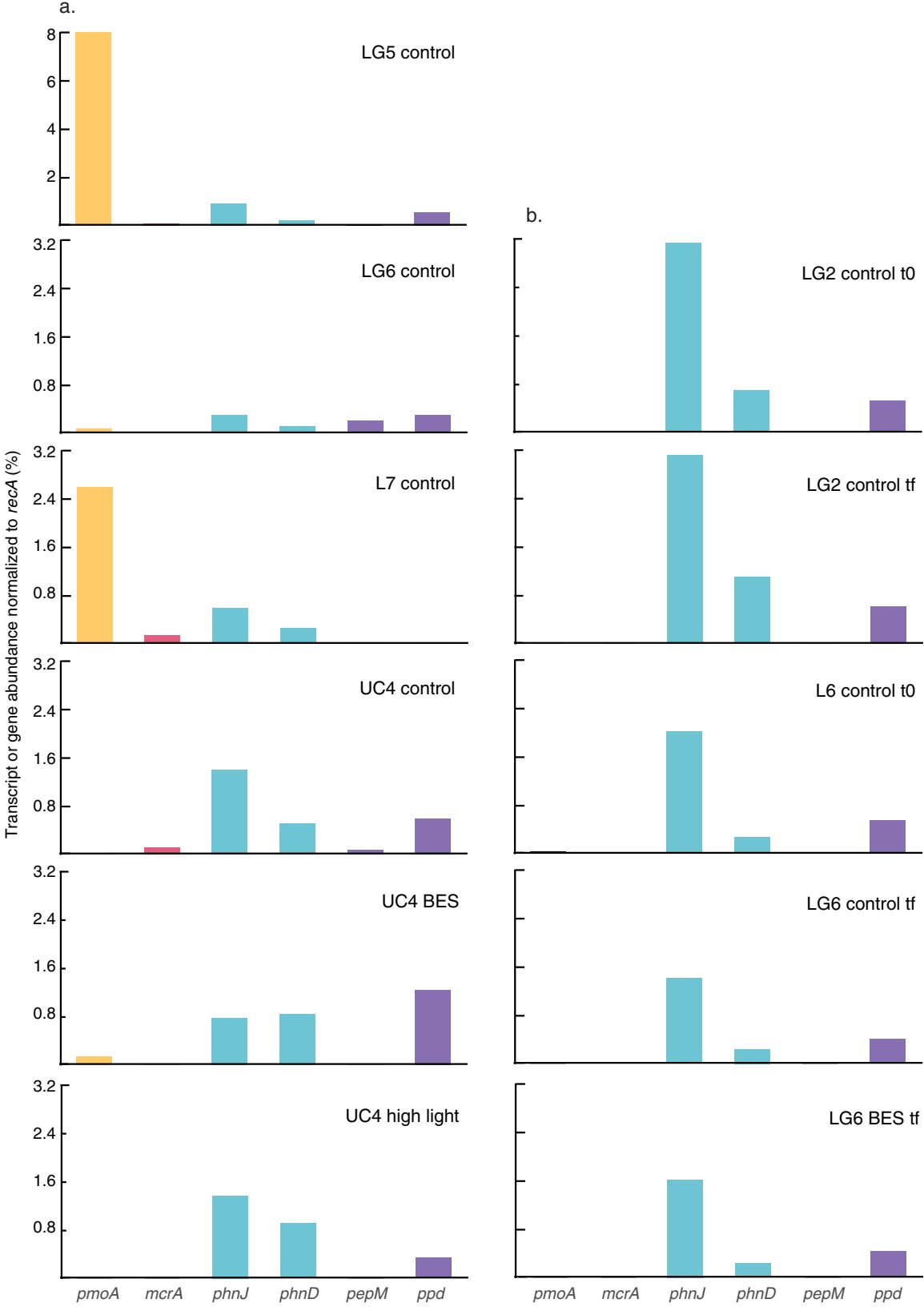

Finally, we assembled and annotated metatranscriptomes and metagenomes to examine whether particular groups possess and express multiple *phn* and related genes in parallel, as the genes involved in phosphonate acquisition and utilization often cluster in genomes[18]. We found that *phnJ* genes commonly clustered with other *phn* genes on the same contigs, particularly those from the *Comamonadaceae* (Table S6). For example, contigs >50 kb containing *phnC*, *phnD*, *phnE*, *phnI*, *phnJ*, *phnK*, *phnL*, *phnM* were present in metagenomes from the LG2 and L6 incubations, and were affiliated with the abundant freshwater bacterial group *Limnohabitans*[30–32], as well as other members of the *Comamonadaceae*, such as *Hydrogenophaga* and *Acidovorax*. Several longer

**Fig. 2 | Functional genes in metatranscriptomes and metagenomes.** Variations across experiments and treatments in (a) transcript abundance (normalized as % of DNA recombination protein (*recA*) transcripts in metatranscriptomes) and (b) gene abundance (normalized as % of *recA* genes in metagenomes) of key functional genes. Functional genes quantified include those involved in: methane oxidation (particulate methane monooxygenase; *pmoA*), methanogenesis (methyl coenzyme M reductase; *mcrA*), phosphonate assimilation (C-P lyase and phosphonate-binding protein; *phnJ* and *phnD*), and phosphonate synthesis (phosphoenolpyruvate mutase and phosphonopyruvate decarboxylase; *pepM* and *ppd*). Abbreviations denote the experiment (e.g., UC4), the treatment (control, addition of the methanogenesis inhibitor 2-bromoethanesulphonate [BES], or high light intensity), and the sampling time point ($t_0$ indicates initial sampling time point and $t_f$ indicates endpoint sample). Note the difference in the vertical axis for the LG5 experiment in comparison with the other experiments. Source data are provided as a Source Data file.

contigs also contained *pst* or *pho* genes involved in P uptake and metabolism (Table S6).

High-elevation lakes in the Sierra Nevada are typically oligotrophic[33–35], and dissolved inorganic phosphorus (DIP) concentrations in the lakes sampled here are consistently near the limits of detection (100 nM) by colorimetric techniques, with the vast majority of measurements <200 nM (refs. 36,37, Table S1). However, Sierra Nevada lakes can also exhibit N-limitation[33], and dissolved inorganic nitrogen (DIN) concentrations in the lakes sampled here are typically <2 µM (Table S1). Under these low dissolved inorganic nutrient concentrations−for both P and N−organic compounds like MPn and MeA could serve as important sources of essential nutrients. Consistent with this, the presence and expression of *phnJ* genes indicates that several microbial groups are capable of releasing $CH_4$ through the cleavage of the C-P bond in MPn (Fig. 2). In particular, our data suggest that multiple genera within the abundant and widely-distributed *Comamonadaceae*[30–32] utilize MPn, given high *phnJ* and 16S identities to sequences collected in other locations (Tables S3 and S5), as well as the recovery of contigs containing *phn* and other genes involved in P acquisition and metabolism (Table S6).

## Evidence for methane production by photosynthetic bacteria

Cyanobacterial photosynthesis was recently identified as a possible additional source of paradoxical $CH_4$ production−yet the pathway by which Cyanobacteria ultimately produce $CH_4$ remains unknown[24]. Based on 16S rRNA genes and transcripts, and metagenomes and metatranscriptomes, Cyanobacteria were common in our samples (Table 1). We used two additional treatments to partition the relative effects of $CH_4$ production vs. oxidation and evaluate Cyanobacteria as a potential source of $CH_4$. First, we included dark treatments in some experiments, as (i) light has been shown to inhibit $CH_4$ oxidation[38–40], and (ii) $CH_4$ production in cyanobacterial cultures was positively correlated with light[24]. As a result, dark treatments would be expected to have higher $CH_4$ oxidation rates, lower cyanobacterial $CH_4$ production, and therefore lower $CH_4$ concentrations compared with controls. Additionally, we increased light intensity in four experiments, which may have a two-fold effect: (i) greater inhibition of $CH_4$ oxidation, and (ii) increased rates of $CH_4$ production from Cyanobacteria or other phytoplankton[8,9,24]. For both of these reasons, we expected to observe higher $CH_4$ production rates under higher light levels.

Dark bottles showed mixed results, with four cases of $CH_4$ consumption, production in two cases, and no significant difference in four experiments (Fig. 1 and S2). In two of four experiments with high light intensity treatments, we observed increased $CH_4$ production (L2 and UC4; both $p < 0.05$). The UC4 experiment was particularly illuminating, as higher light intensity significantly boosted $CH_4$ production rates by 62-fold compared with the control (up to 3.2 nM h$^{-1}$; $P < 0.0005$). This was among the strongest treatment effects observed across all experiments (Fig. S2). BES also increased the $CH_4$ production rate by 49-fold compared with control in the UC4 experiment (up to 2.5 nM h$^{-1}$; $P < 0.0005$, Fig. S2). This counterintuitive effect of increasing $CH_4$ production−despite the addition of a known methanogenesis inhibitor−suggests another source of $CH_4$.

To examine these responses in greater detail, we analyzed both the $\delta^{13}C$ stable isotope composition of $CH_4$ produced during the UC4 experiment, as well as changes in gene expression in response to experimental treatments in metatranscriptomes. We directly compared these results with the earlier UC3 experiment−which also showed $CH_4$ production in the BES treatment (there was no high light treatment)−and with the LG6 and L7 experiments−which did not show $CH_4$ production in BES or high light treatments. Consistent with a lack of significant $CH_4$ production in the LG6 and L7 experiments, $\delta^{13}CH_4$ from multiple sampling time points showed no significant variation between treatments in the LG6 and L7 experiments (Fig. 3a, b).

UC4 experimental data presented a clear contrast, as $\delta^{13}CH_4$ values were significantly lower in the BES and high light treatments compared with controls (Fig. 3c). $\delta^{13}CH_4$ values in the UC3 BES treatment showed similar values and also differed from the control (Fig. 3c). Along with increased $CH_4$ concentrations (Fig. 3d), these data are indicative of an isotopically depleted $CH_4$ source in the BES and high light treatments, which must be <−55‰ based on isotopic mass balance. This value depends on the rate at which $CH_4$ is oxidized, which acts to isotopically enrich the remaining $CH_4$ pool[15]. As a result, higher $CH_4$ oxidation rates would imply an even more $^{13}C$-depleted source of $CH_4$. However, $CH_4$ oxidation was likely minimal given that $\delta^{13}CH_4$ and $CH_4$ concentrations showed little change in control incubations, as well as the fact that high light intensity can inhibit $CH_4$ oxidation[38–40].

Isotopically depleted $\delta^{13}CH_4$ values of <−55‰ associated with $CH_4$ production were substantially lower than atmospheric $CH_4$ samples collected at the same time (−45.2 to −47.4‰), and reflect isotopic fractionation by the process producing $CH_4$ (ref. 41). Methanogenesis does lead to strong fractionation, but our measured values fall at the upper end of $\delta^{13}CH_4$ values for methanogenesis (−110 to −50‰), consistent only with acetoclastic methanogenesis[41]. However, *mcrA* expression was undetectable (Fig. 2) in these treatments−which include BES, a known methanogenesis inhibitor−suggesting another source. In order for MPn metabolism to produce $\delta^{13}CH_4$ values of ca. −55‰, MPn would need to have a similar isotopic composition, as the mean isotopic fractionation for $CH_4$ derived from MPn is only 1.3‰ (ref. 42). MPn could be allochthonous (terrestrial) or autochthonous (based on *pepM* results; Fig. 2), but in either case, seems unlikely to be sufficiently $^{13}C$-depleted. The C atom in MPn is derived from the intramolecular rearrangement of phosphoenolpyruvate (PEP; refs. 17,18), and PEP undergoes continuous intracellular regeneration; within organisms producing MPn, PEP would need to be $^{13}C$-depleted by at least 20-30‰ compared with overall cellular C in order to yield $\delta^{13}C$ values of ca. −55‰ for MPn. Instead, $\delta^{13}CH_4$ data suggest another paradoxical $CH_4$ production mechanism, and similar $\delta^{13}CH_4$ values have been interpreted as evidence for $CH_4$ production by photosynthetic organisms[43].

Although the pathway by which Cyanobacteria produce $CH_4$ remains unidentified, Bižić et al.[24] hypothesized that $CH_4$ is produced during photosynthesis owing to positive correlation between light and $CH_4$. We used metatranscriptomic data to identify specific functions that were significantly different in the treatments vs. the control, and that may explain differences in $CH_4$ concentrations and isotopic composition. Through comparison of all functional families in metatranscriptomes from the UC4 experiment[44], we identified two functions that differed strongly and significantly between the control and the BES ($P < 0.0005$) and high light intensity ($P < 0.005$) treatments (Fig. 4; Supplementary Note 2). Interestingly, both were related to chlorophyll biosynthesis; ferredoxin:protochlorophyllide reductase (DPOR) and chlorophyllide a reductase (COR) are involved in porphyrin and chlorophyll metabolism. DPOR is found in photosynthetic

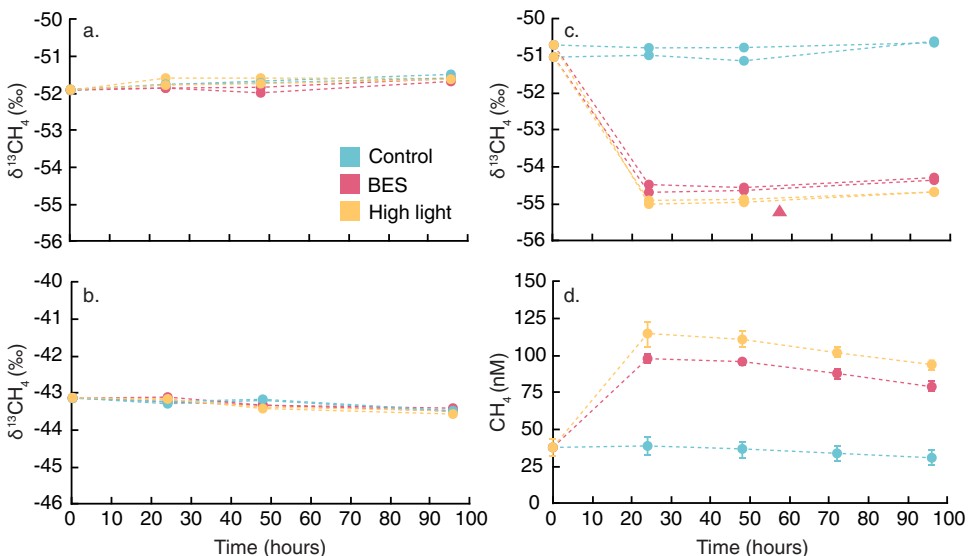

**Fig. 3 | δ¹³CH₄ and CH₄ in experimental treatments.** $\delta^{13}CH_4$ values are shown over time for duplicate samples collected during the (a) L7, (b) LG6, and c UC4 and UC3 experiments. d CH₄ concentrations are shown over time in the UC4 experiment. Different colors denote different experimental treatments and controls (BES indicates the addition of the methanogenesis inhibitor 2-bromoethanesulphonate).

Scales of the vertical axes differ between experiments in **a**–**c**. In (**c**), the triangle denotes the UC3 BES treatment; controls are not shown due to a lack of change. In **d**, error bars denote the standard deviation of the mean of triplicate concentration measurements; in some cases, these are smaller than the datapoints and are not visible. Source data are provided as a Source Data file.

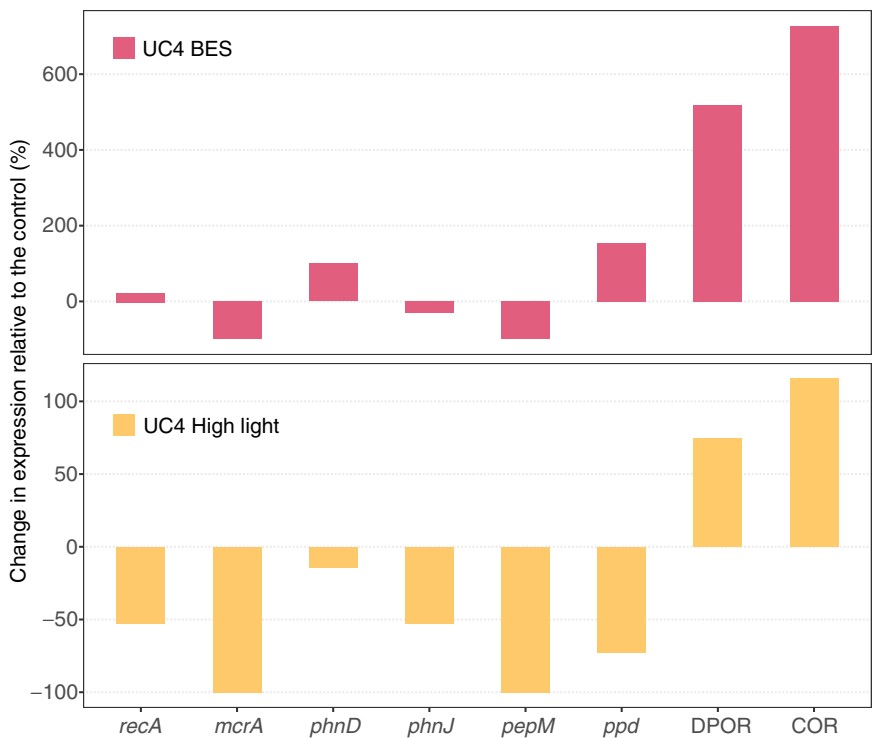

**Fig. 4 | Gene expression in the UC4 experimental treatments compared with the control.** Functional genes involved in methanogenesis (methyl coenzyme M reductase; *mcrA*), phosphonate assimilation (phosphonate-binding protein and C-P lyase; *phnD* and *phnJ*), phosphonate synthesis (phosphoenolpyruvate mutase and phosphonopyruvate decarboxylase; *pepM* and *ppd*), and porphyrin and chlorophyll metabolism (ferredoxin:protochlorophyllide reductase and chlorophyllide

reductase; DPOR and COR) are shown. The BES treatment (addition of the methanogenesis inhibitor 2-bromoethanesulphonate) is shown in the top panel and the high light treatment in the bottom panel. Each panel shows the net effect of the treatment on the different genes in metatranscriptomes. DNA recombination protein (*recA*) transcripts are also shown for comparison. Source data are provided as a Source Data file.

bacteria, Cyanobacteria, and green algae, and is involved in the light-independent reduction of protochlorophyllide[45,46]. COR catalyzes the first step in the conversion of chlorin to a bacteriochlorin ring during bacteriochlorophyll biosynthesis[47–49]. Notably, both DPOR and COR are nitrogenase-like enzymes, and both were abundant in all

metatranscriptomes and metagenomes (Fig. 5). DPOR was found in several cyanobacterial groups—such as *Pseudanabaena*, *Dolichospermum*, *Snowella* and *Oscillatoria*—in the metatranscriptome data (Table S3). However, both DPOR and COR were most commonly found within the *Limnohabitans* and *Polynucleobacter* genera (36.3–54.5% of

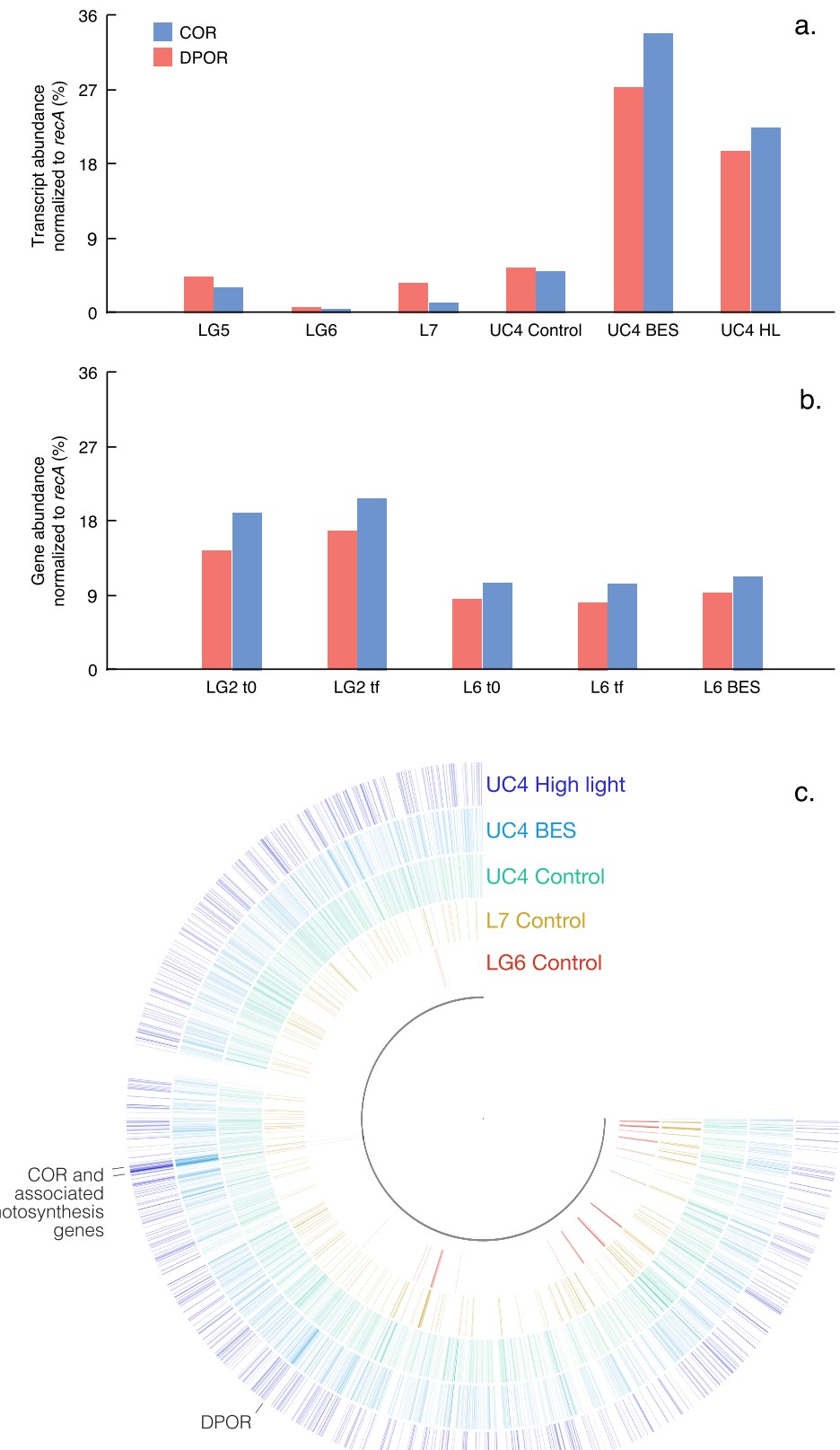

DPOR and COR transcripts; Table S3), as some strains of these abundant freshwater bacteria are capable of aerobic anoxygenic photosynthesis[32,50].

To further verify these findings, we mapped reads from the UC4 experiment, as well as the LG6 and L7 experiments (which did not show $CH_4$ production), to metagenome-assembled genomes (MAGs). 27 MAGs were assembled from two Yosemite samples as part of a global-scale analysis[51], and five of these MAGs were betaproteobacterial. This included one *Rubrivivax* MAG, one *Polynucleobacter* MAG, and three *Limnohabitans* MAGs. All five of these betaproteobacterial MAGs contained genes for photosynthesis, including COR (Table S7). Three of these MAGs contained both COR and DPOR, as well as multiple

**Fig. 5 | Expression and abundance of bacterial photosynthesis genes.** Protochlorophyllide reductase (DPOR) and chlorophyllide reductase (COR) reads in **a** metatranscriptomes, (**b**) metagenomes, and **c** mapped to the *Limnohabitans* metagenome-assembled genome (MAG) 5. In **a**, **b**, DPOR (red) and COR (blue) transcripts (**a**) and genes (**b**) are shown within different experiments and experimental treatments. Abbreviations denote the experiment (e.g., UC4), the sampling time point, and the treatment (control, addition of the methanogenesis inhibitor 2-bromoethanesulphonate [BES], or high light intensity). In **c**, the *Limnohabitans* MAG 5 is shown in circular form, and the number of reads mapping to genes within the MAG are indicated by color intensity. Different experiments and treatments are shown in different colors, with the LG6 experiment in red, L7 in orange, the UC4 control in aqua, the UC4 BES treatment in light blue, and the UC4 high light treatment in dark blue. Locations of DPOR and COR (and associated photosynthesis genes) within the MAG are indicated. Source data are provided as a Source Data file.

genes involved in phosphonate metabolism. Of the remaining two MAGs, one lacked DPOR while the other lacked *phn* genes (Table S7)—but these MAGs were estimated to be 55–58% complete. Interestingly, the Yosemite *Limnohabitans* MAG 5 shared 95% average nucleotide identity (ANI) to several MAGs assembled from Canadian Lakes where paradoxical $CH_4$ production was previously observed[9,43].

Based on read mapping to these MAGs, we examined patterns in transcripts that showed differences in the BES and high light treatments of the UC4 experiment compared with the control. We also examined differences between the UC4 experiment versus the LG6 and L7 experiments. We found that multiple photosynthesis genes with high similarity to *Limnohabitans* MAG 5 were highly expressed in UC4, while not detected at all in the LG6 and L7 experiments (Fig. 5c). Furthermore, of >12,000 genes present in the five MAGs, several showed the highest overall number of mapped reads in the UC4 BES and high light treatments. Principal among these were two subunits of COR, as well as associated photosynthetic reaction center genes located on the same MAG scaffold (Fig. 5c). Transcripts with high similarity to the *Limnohabitans* MAG 5 DPOR gene were also highly expressed. In contrast, phosphonate metabolism genes were expressed at lower levels and in similar levels in all metatranscriptomes.

DPOR and COR were expressed and present in all metatranscriptomes and metagenomes, yet the overall expression of DPOR and COR was an order of magnitude higher in the UC4 experiment compared with the LG6 and L7 experiments (Fig. 5a). Differences in the expression of DPOR and COR from *Limnohabitans* MAG 5 were even more pronounced. These patterns are consistent with the production of $CH_4$ and changes in $\delta^{13}CH_4$ that were observed in the UC4 experimental treatments, and which were not observed in the LG6 and L7 experiments (Fig. 3). Based on the positive effects of BES and of high light intensity on $CH_4$ production observed in the UC4 experiment, this mechanism may also be active in the L2, L3, LG4, and UC3 experiments, where BES or high light intensity also significantly increased $CH_4$ production (Fig. S2). Decreases in $CH_4$ in dark treatments in several experiments may also reflect decreased photosynthesis (alongside increased $CH_4$ oxidation; Fig. S2). Experimental data further indicate that this mechanism is environmentally variable, given clear differences between experiments (Figs. 3 and 5). However, these variations are consistent with the fact that photosynthesis by *Limnohabitans* is variable, as photosynthesis is used to supplement heterotrophy[50].

Based on these findings, we propose two potential mechanisms that could be involved in paradoxical $CH_4$ production in oxic surface waters of freshwater lakes: (i) $CH_4$ may be produced from methoxyl groups present in (bacterio)chlorophyll precursors, or (ii) $CH_4$ production may be catalyzed by DPOR and COR enzymes. In terrestrial plants, $CH_4$ is thought to be produced aerobically from structural components—such as pectin, lignin and cellulose[52–54]—particularly in plants under stress (such as increased temperature, UV radiation, or physical damage[55]). Protochlorophyllide, chlorophyllide, and other (bacterio)chlorophyll precursors contain methoxyl groups that have been shown to serve as precursors of $CH_4$ in plants[56], and a similar mechanism may be active in Cyanobacteria and/or Proteobacteria under stress (presumably due to BES additions and high light intensity in our experiments). Alternatively, $CH_4$ production may be catalyzed by DPOR and COR enzymes. Nitrogenases reduce a range of multi-bond compounds[57–61], and this quality is shared across different

nitrogenases[62]. Similar to the findings of Zheng et al.[60], the nitrogenase-like enzymes DPOR and COR may be able to catalyze the reduction of carbon dioxide ($CO_2$) into $CH_4$. Importantly, DPOR and COR are central in (bacterio)chlorophyll metabolism and photosynthesis, and so DPOR is present in Cyanobacteria, while both are found within the abundant and ubiquitous freshwater bacterial groups *Limnohabitans* and *Polynucleobacter*[30–32,50].

## Discussion

Our combined results indicate that a novel paradoxical $CH_4$ production mechanism associated with photosynthesis co-occurs with other mechanisms in aquatic ecosystems, with several implications for our understanding of aquatic $CH_4$ production. First, confirmation of the $CH_4$ paradox in freshwater is still limited in scope and relatively recent[8,9,15,24,25], yet freshwater lakes are large sources of $CH_4$ to the atmosphere[2]. We conducted multiple experiments in multiple lakes, and measured significant $CH_4$ production in controls and/or treatments in 63% of experiments (Fig. 1). In many of the remaining experiments, variations in $CH_4$ concentrations over time, across experimental treatments, and in *pmoA* gene expression patterns suggest that $CH_4$ oxidation occurs in parallel with $CH_4$ production (Fig. 2, SI Text). Variation in $CH_4$ oxidation relative to production may be one source of the variability that we observed across experiments. $CH_4$ oxidation also complicates analysis of paradoxical $CH_4$ production because it may obscure $CH_4$ production. As a result, use of different experimental treatments combined with 'omic data, or other approaches, are required to identify and examine paradoxical $CH_4$ production in greater detail.

Given the ubiquity of *phnJ* genes and transcripts in our data (Fig. 2), their presence on contigs from abundant organisms (Table S6), their similarity to sequences collected in other locations (Tables S3 and S5), and the potential for P limitation in freshwater ecosystems, microbial metabolism of MPn may represent an important baseline $CH_4$ production mechanism in freshwater lakes. This fits with current understanding of marine ecosystems[6,16]. Our data further demonstrate that MPn may itself be produced within lakes based on *ppd* and *pepM* expression (Fig. 2). Quantifying the relative contributions of allochthonous vs. autochthonous MPn production—as well as broader P acquisition dynamics—is therefore essential to understanding $CH_4$ production in freshwater. In the lakes studied here, MPn may be more widely available where dissolved organic carbon (DOC) concentrations are higher in Lukens Lake (Table S1). However, further organic matter characterization[16] is needed to determine relative MPn concentrations in freshwater through space and time. Phosphonate compounds were once thought to be resistant to breakdown, and their formation remains poorly understood;[17,63] variation in their production and availability in comparison with other forms of P and N may drive variations in $CH_4$ production such as those evident in our data.

Inorganic P and especially nitrogen (N) may be supplied through atmospheric deposition and snowmelt (particularly at higher elevations)[33,64], for instance, influencing overall DIP and DIN availability, and the degree of N versus P limitation[33]. Our metatranscriptomic data indicate that both MPn and MeA metabolism may occur under the low nutrient conditions prevalent in high elevation lakes. This is consistent with the fact that both MPn and MeA are used in Yellowstone Lake[23], and provides evidence that MeA metabolism and associated paradoxical $CH_4$ production occurs in additional

freshwater bodies. However, *Caulobacter aat* sequences were more prevalent in Yosemite—suggesting that different bacterial groups may be relevant for MeA metabolism in different locations or under different environmental conditions. Finally, our data and those of Wang et al.[23] raise the possibility that *Polaromonas* and *Limnohabitans* also metabolize MeA.

Photosynthesis represents an additional, newly-identified, and potentially widespread $CH_4$ source that occurs via an unknown pathway[24]. As a consequence, we used a combination of approaches to identify whether photosynthetic $CH_4$ production occurs in freshwater, and by what means. Although independent data types ($CH_4$ concentrations, treatment effects, isotopic data, 'omic data) may be interpreted in multiple ways, the combined data are consistent with photosynthetic $CH_4$ production (Figs. 3 and 4). For example, high light intensity and BES treatments boosted $CH_4$ concentrations (Fig. S2), but this $CH_4$ is unlikely to result from methanogenesis given scant *mcrA* expression (Fig. 4) and the use of a known methanogenesis inhibitor. Instead, DPOR and COR transcripts were present and responsive to treatments as shown by metagenomic and metatranscriptomic data. $\delta^{13}CH_4$ data are further indicative of a $CH_4$ source that falls at the uppermost end of possible values for (acetoclastic) methanogenesis[41], or requires a substantially $^{13}C$-depleted pool of PEP/MPn[17,18,42], but are most consistent with an interpretation of photosynthetic $CH_4$ production[43]. Extending from the UC4 experiment, photosynthetic $CH_4$ production may take place in at least four other experiments where BES and high light intensity also significantly increased $CH_4$ (L2, L3, LG4, UC3; Fig. S2). Although BES and high light intensity produced surprisingly similar patterns in $CH_4$ production, $\delta^{13}CH_4$ values, and gene expression (Figs. 3–5), there are several possible explanations for why BES may induce a response (Supplementary Note 3). Finally, the ubiquity of DPOR and COR genes and transcripts indicates that the potential for photosynthetic $CH_4$ production is widespread (Fig. 5; Tables S3 and S7), yet the apparent variability across experiments is consistent with known variability in aerobic anoxygenic photosynthesis by *Limnohabitans*[50,65]. In particular, aerobic anoxygenic photosynthetic activity is dependent upon the concentrations and forms of available organic matter, light levels, the prevalence of particular bacterial groups, and interactions between these factors[66–69]—which may explain some of the variations observed across our experiments. High light treatments in particular indicate that photosynthetic $CH_4$ production is variable, as elevated light levels had no effect in some experiments and strong effects in others. Ultimately, periodic pulses of photosynthetic $CH_4$ production may be superimposed on top of consistent production from MPn or MeA (under nutrient-limiting conditions), with rare contributions from water column-based methanogenesis. $CH_4$ concentrations and fluxes may be subsequently affected by $CH_4$ oxidation, leading to additional variation seen here and in other studies. Research in additional lakes should examine whether this overall pattern of $CH_4$ sources and sinks holds elsewhere.

Photosynthetic $CH_4$ production was most clearly evident in experimental treatments, highlighting the efficacy of including different treatments to examine paradoxical $CH_4$ production. We leveraged data from these treatments to identify potential $CH_4$ production mechanisms—providing an experimental approach, isotopic signature data, and two potential gene targets to examine in other aquatic ecosystems. We found that this mechanism is not limited to the Cyanobacteria, as the majority of DPOR and COR transcripts and genes were derived from the betaproteobacteria. This has several potential implications. First, *Limnohabitans* and *Polynucleobacter* are ubiquitous in freshwater ecosystems[30,31]. MAGs from both of these groups contained DPOR and COR genes, as well as genes for phosphonate metabolism. *Limnohabitans* also expressed *phnJ* genes, indicating that they can play a dual role in paradoxical $CH_4$ production. These findings suggest that two possible $CH_4$ production mechanisms (photosynthetic and phosphonate-based) co-occur in multiple lineages of

abundant and widespread bacteria present in freshwater. Additional work with isolates and environmental samples should determine the degree to which these mechanisms are active, as well as how different environmental factors (such as light, different dissolved compounds, and other relevant factors) influence $CH_4$ production. Second, our data are indicative of an additional paradoxical $CH_4$ production mechanism—$CH_4$ production by aerobic anoxygenic photosynthetic (AAnP) bacteria—that may also be relevant in the ocean. AAnP bacteria constitute up to 10% of marine bacterial communities and play an important role in ocean carbon cycling[65]. Given their significance, the potential for $CH_4$ production by marine AAnP deserves examination. Finally, multiple other nitrogenase-like enzymes produce $CH_4$ (refs. 60,62), and our results implicate the nitrogenase-like enzymes DPOR and COR. Whether these underpin $CH_4$ production in other photosynthetic organisms remains to be determined. Our combined dataset indicates that this mechanism may produce $CH_4$ paradoxically, is present alongside other mechanisms (even within the same organisms), and is likely widely-distributed in aquatic ecosystems—representing another potentially important source of $CH_4$ to the atmosphere.

## Methods

### Field sites, sample collection, and experimental set-up

Water samples were collected in 2016–2018 in five high-elevation lakes in Yosemite National Park, and used in incubation experiments. Samples were collected in Lukens (L), Lower Cathedral (LC), Upper Cathedral (UC), Lower Gaylor (LG), and Upper Gaylor (UG) Lakes. Experiments are denoted by lake abbreviation and sequential numbering for each lake (Table S2). Water samples were collected in the littoral and limnetic zones of the lakes at 0.1 m depth with acid-washed cubitainers, and then kept on ice or refrigeration to maintain in-lake temperature until laboratory incubations were established within 24 h of sample collection. Temperature and dissolved oxygen were measured at the time of sample collection using a ProODO YSI probe (YSI Inc., Yellow Springs, OH, USA).

Water collected in the lakes was transferred to 170 or 300 ml Wheaton bottles, capped and crimped, and a known volume of air was introduced to generate a headspace for sampling. Initial $CH_4$ samples were collected, and bottles were incubated in water baths at the temperature at which water samples were collected in the field. All controls and experimental treatments were conducted in triplicate, with three bottles sampled sacrificially at each measurement timepoint during the experiments. The different treatments used in the experiments were:

- Control: unamended lake water following a natural day-night set up (water bath lid was opened at 7:00 h and closed at 18:00 h).
- Dark: unamended lake water; bottles were kept in the dark during the whole incubation time.
- BES: lake water was amended with 2-bromoethanesulphonate (BES) to a final concentration of $5 \times 10^{-4}$ M. This concentration has been established to inhibit methanogenesis[27] and has been used in methane paradox experimentation previously[8]. This treatment followed the same natural daylight set-up as the control.
- High light: unamended lake water was subjected to 500 μmol $m^{-2} s^{-1}$ inside a growth chamber and followed the same natural daylight set-up as the control (light would turn on at 7:00 h and off at 18:00 h).
- Sterile treatments: 0.22 μm-filtered lake water did not show significant $CH_4$ production or oxidation over time, indicating observed $CH_4$ production and consumption is likely biotic.

For each of these, three bottles were sampled sacrificially every 6–24 h for up to 96 h by collecting gas samples from the headspace with a syringe, and immediate transfer to Exetainers (Labco Ltd.,

Lampeter, Ceredigion, UK) for later analyses. Not all treatments were tested in each incubation experiment. Temperature and oxygen concentrations were monitored at each sampling point. Optical sensor spots (Fibox, Loligo Systems, Viborg, Denmark) were used to measure oxygen concentrations during incubations (detection limit of 100 nM) and ensure that the water did not go anoxic at any time. All measured values exceeded 5.2 mg L$^{-1}$ (Table S1). The temperature was measured with the Fibox temperature sensor and kept constant during the incubation time. Water samples were filtered at the beginning and end of the incubation for DNA and/or RNA sampling.

## Methane measurements

Methane concentrations were measured via headspace equilibration and gas chromatography. Headspace gas samples from incubations were collected with a gas-tight syringe into 12-mL Labco Exetainer vials (Labco Ltd., Lampeter, Ceredigion, UK) after incubation bottles were shaken for 2 min to reach equilibration. Samples were subsequently analyzed using a Shimadzu GC-2014 gas chromatograph (Shimadzu, Kyoto, Japan) with flame ionization detection (FID) for CH$_4$ (ref. [70]). Headspace CH$_4$ concentration measurements were then used to calculate CH$_4$ concentration in lake water based on Henry's law of equilibrium[71].

## DNA and RNA Extraction

Water samples were filtered through 0.22 μm filters (Millipore, Darmstadt, Germany), and DNA filter samples were preserved in Sucrose-Tris-EDTA (STE) buffer in pre-prepped Lysis Matrix E tubes (MP Bio, Eschwege, Germany) and frozen at −80 °C until extraction. RNA samples were preserved in RNA*later*® (Ambion™, AM7021) in pre-prepped Lysing Matrix E tubes and frozen at −80 °C until extraction. DNA was extracted using the Qiagen DNeasy Blood & Tissue Kit with a modified protocol from Beman et al.[72]. Briefly, samples were lysed with 100 μL 10% sodium dodecyl sulfate (SDS) and DNA was separated from proteins and cellular debris using proteinase K (20 mg mL$^{-1}$; Qiagen, Inc., Valencia, CA, USA); then DNA was precipitated with ethanol and purified following the manufacturer's instructions. After extraction, samples were preserved at −80 °C until further analyses. RNA was extracted using a *mir*Vana miRNA Isolation Kit (Ambion™, AM1560) with a modified protocol from Huber and Fortunato[73]. Briefly, samples were lysed with the kit's lysing matrix, then subjected to an organic extraction with phenol-chloroform, followed by a wash to obtain RNA. Immediately after RNA extraction, we used the SuperScript III First-Strand Synthesis System (Life Technologies Corporation, Carlsbad, CA, USA) to synthesize first-strand cDNA, and samples were preserved at −80 °C until further analyses. DNA, RNA, and cDNA purity was measured using a Biospectrometer (Eppendorf AG, Hamburg, Germany) and the concentrations were quantified using PicoGreen Quant-iT dsDNA quantitation assay (ThermoFisher Scientific, USA) for DNA samples and the MaestroNano Pro (Maestrogen Inc., Taiwan) for RNA and cDNA samples.

## 16S sequencing

DNA and cDNA extracted from filtered water samples were diluted to a common concentration (1 ng μl$^{-1}$) and sent for 16S rRNA amplicon sequencing on an Illumina MiSeq (Illumina, San Diego, CA, USA) according to Earth Microbiome protocols. We used the universal primers 515F-Y (5′-GTGYCAGCMGCCGCGGTAA) and 926R (5′-CCGYCAATTYMTTTRAGTTT). DNA samples were sequenced at the Joint Genome Institute (Berkeley, CA, USA) and cDNA samples at the Argonne National Laboratory (Lemont, IL, USA).

ASVs were generated from 16S rDNA and rRNA sequence data using the Divisive

Amplicon Denoising Algorithm (DADA2) v1.12 (ref. [74]) as implemented in QIIME 2 v2019.7.0 (ref. [75]), and then used for subsequent analyses. After import and demultiplexing, read quality was visualized

using the 'qiime tools view' command. Reads were then processed using the 'qiime dada2 denoise-paired' command, with 13 bp trimmed from both the forward and reverse reads, truncation of reverse reads to 169 bp (due to the well-known decline in sequence quality observed for MiSeq reverse reads), and training of the denoising algorithm on 1 million reads. Classification of ASVs was conducted in mothur v1.42.2 (ref. [76]) using the SILVA (version 128) database.

## Metatranscriptomics and metagenomics

Metatranscriptomes and metagenomes were generated from individual treatments in selected experiments in order to examine potential production mechanisms and coupled methane oxidation. Following extraction, DNA and RNA samples were sent for sequencing in the Vincent J. Coates Genome Sequencing Laboratory (GSL) at the University of California, Berkeley (https://genomics.qb3.berkeley.edu/), which is supported by NIH S10 OD018174 Instrumentation Grant. For each DNA sample, 250 ng of genomic DNA was sheared and libraries were prepared using the KAPA HyperPrep Kit (Kapa Biosystems, Wilmington, MA, USA). For each RNA sample, -800 ng of total RNA was depleted of rRNA using the Ribo-Zero rRNA Removal Kit (Illumina, Inc., San Diego, CA, USA), sheared, and libraries were prepared using the KAPA RNA HyperPrep Kit (Kapa Biosystems, Wilmington, MA, USA). Twelve samples were pooled into a single lane and sequenced via 150-cycle paired-end sequencing on the Illumina HiSeq 4000 platform (Illumina, Inc., San Diego, CA, USA).

Data were demultiplexed by the GSL and reads were filtered and trimmed using BBDuk v38.23 (https://jgi.doe.gov/data-and-tools/software-tools/bbtools/bb-tools-user-guide/bbduk-guide/) with the following parameters: maq = 8, maxns = 1, minlen = 40, minlenfraction = 0.6, k = 23, hdist = 1, trimq = 12, qtrim = rl. Forward and reverse reads were then merged using PANDASeq v2.11 (https://github.com/neufeld/pandaseq; ref. [77]) with default parameters. Merged reads were subsequently queried against the NCBI NR database (accessed February 11$^{th}$, 2020 and including Subject Taxonomy IDs) using the BLASTX function in DIAMOND v0.9.30 (http://diamondsearch.org/; ref. [78]) with the following search criteria: maximum number of target sequences = 1, bit-score >40. In order to quantify functional gene abundances, we filtered BLASTX matches to functional genes of interest based on the DIAMOND-generated text annotation for each metagenome or metatrascriptome, discarding those with similarity below 60%. We normalized the abundance of DNA recombination protein (*recA*) genes/transcripts in each metagenome or metatranscriptome. Metagenomic contigs were assembled via megahit v1.1.3 (ref. [79]) using an initial k-mer size of 23, and were annotated as described above but with the 'long reads' option in DIAMOND.

For additional analyses, UC4 metatranscriptomes were uploaded to the Joint Genome Institute (JGI; https://img.jgi.doe.gov/) Integrated Microbial Genomes & Microbiomes platform (IMG/M). We used IMG/M's comparative analysis tools to compare the functional capabilities between the UC4 experiment metatranscriptomes[44]. The 'abundance profile viewer' tool was used to provide an overview of the relative abundance of the functional families across the UC4 control vs. treatment metatranscriptomes, and the 'abundance comparisons' tool was used to test whether the differences in the abundance of specific functions (across the UC4 control vs. treatment metatranscriptomes) were significant. We also used the phylogenetic distribution tool to quantify the abundance of different phylogenetic groups of potential importance in the methane paradox.

Metatranscriptome reads were also mapped to MAGs using bowtie2 v2.3.4.3 (ref. [80]) and visualized using Anvi'o (ref. [81]). Coverage files were generated in samtools v1.11 (http://www.htslib.org/) for statistical comparisons subsequently conducted in R. MAGs were assembled during the global analysis by Nayfach et al.[51]. In brief,

assembled metagenomes in IMG/M were binned into MAGs using MetaBAT[82] on the basis of tetranucleotide frequencies using v0.32.4 and 0.32.5 with option '--superspecific.' MAGs were refined using RefineM v0.0.20 (ref. 83) to remove contigs with aberrant read depth, GC content, and/or tetranucleotide frequencies, and by removing contigs with conflicting phylum-level taxonomy (based on protein-level alignments against the IMG/M database using Last v876; ref. 84). Finally, CheckM v1.0.11 (ref. 85) was used to select MAGs that were at least 50% complete, with less than 5% contamination, and with a quality score >50.

### Stable isotopic measurements

Headspace of incubation bottles was transferred to evacuated exetainers and sent to the UC Davis Stable Isotope Facility (Davis, CA) for analysis. Measurements of stable isotope ratios of carbon ($\delta^{13}C$) in $CH_4$ were conducted using a ThermoScientific Precon concentration unit interfaced to a ThermoScientific Delta V Plus isotope ratio mass spectrometer (ThermoScientific, Bremen, Germany). In brief, gas samples are passed through a $H_2O/CO_2$ scrubber and a cold trap, and $CH_4$ is then separated from other gases and oxidized to $CO_2$. Pure $CO_2$ reference gas is used to calculate provisional $\delta$ values, and final $\delta$ values are calculated after correcting for changes in linearity and instrumental drift, and expressed relative to the Vienna PeeDee Belemnite (V-PDB) standard.

### Statistical analyses

All statistical analyses were conducted in the R statistical environment (RStudioVersion 1.2.5001). Pearson correlation was used to assess the relationship between time and $CH_4$ concentration in all incubations and across all replicated treatments. We used the 'lm' function in R, and a priori significance level was defined as $\alpha < 0.05$. A significantly positive relationship between time and $CH_4$ was considered as net $CH_4$ production in the incubation, whereas a significantly negative relationship was considered as net $CH_4$ consumption. The slope of these relationships and their standard errors are reported in Fig. 1.

We subsequently used the 'segmented' package v1.3-4 in R to perform piecewise regression to test for cases where $CH_4$ production or oxidation varied significantly between sampling time points, resulting in nonlinear patterns in $CH_4$ concentrations over the course of the experiment. For example, initial $CH_4$ production could be followed by subsequent oxidation, resulting in significant $CH_4$ increases followed by significant decreases.

To express the different responses in terms of $CH_4$ concentration over time among the different experiment treatments we first calculated the response ratios (lnRR) by the following equation:

$$\text{lnRR} = \ln \frac{\text{mean } CH_4 \text{ in treatment}}{\text{mean } CH_4 \text{ in control}} \qquad (1)$$

Additionally, we calculated which treatments were significantly different from each other and the control using ANOVA ('aov' function) and Tukey honestly significant difference (HSD) post-hoc tests ('HSD.test' function in the agricolae package v1.3.3).

### Reporting summary

Further information on research design is available in the Nature Research Reporting Summary linked to this article.

## Data availability

DNA and RNA sequence data from this study are available in the Sequence Read Archive under BioProject PRJNA656136. $CH_4$ concentration data (https://doi.org/10.6071/M3J67R) and $\delta^{13}CH_4$ data (https://doi.org/10.6071/M3NX08) are available in Dryad. Source data are provided with this paper.

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

## Acknowledgements

This work was supported by the University of California through the Valentine Eastern Sierra Reserve graduate student research grant (to EPC), the Institute for the Study of Ecological and Evolutionary Climate Impacts fellowship (to EPC), University of California Merced Environmental Systems summer grants (to EPC), and the UC MEXUS-CONACYT doctoral fellowship (to EPC). Metagenome sequencing was partially supported by the United States Department of Energy Joint Genome Institute Community Sequencing Program (CSP 1839 to JMB). We thank the United States National Park Service for facilitating sample collection in Yosemite National Park under permits YOSE-2016-SCI-0118, YOSE-2017-SCI-0104, and YOSE-2018-SCI-0091. We thank Steve Hart for his help processing methane samples; Jay Sexton for his help with incubation chambers; and Angela Yu, Joaquin Fraga, Sonia Vargas, Ariadna Cairo, Samantha Vazquez, Jorge Montiel and Daniela Alonso for their help with field and laboratory work.

## Author contributions

Conceptualization: E.P.C., J.M.B.; Methodology: E.P.C., J.M.B.; Investigation: E.P.C., J.M.B.; Visualization: E.P.C., J.M.B.; Supervision: J.M.B.; Writing—original draft: E.P.C.; Writing—review & editing: J.M.B., E.P.C.

## Competing interests

The authors declare no competing interests.
