## [Peer Review File · Nature Communications]

Multiple sources of aerobic methane production in aquatic ecosystems include bacterial photosynthesisReviewer #1 (Remarks to the Author):

The authors used a combination of incubation experiments and 'omic' analysis to study the phenomenon of oxic-water methane production (OWMP). They concluded that the production was functionally linked to bacterial chlorophyll and photosynthesis, and taxonomically to cyanobacteria and proteobacteria.

OWMP contradicts the prevailing paradigm of biological methanogenesis being a strictly anaerobic process, and it may explain the widely reported 'methane paradox' in aquatic systems. Because the idea goes against the 'text book' knowledge about methane cycle, it has been hotly contested--- not unusual for any ground-breaking idea in science. Nevertheless, the argument for OWMP will only strengthen when researchers continue to identify the organisms and biochemical mechanisms that underpin the phenomenon. In this regard, this paper offers some important and new findings. The experiments appear to be well executed. The use of omic techniques produces useful insights into the responsible organisms.

Overall, it is a welcomed contribution to OWMP research, but I find the paper a bit frustrating to read at times when the key messages are buried in the rather tedious description of the different observations in the different lakes and the different treatments. If the authors can improve the organization and presentation of the key findings, it would help readers to grasp the importance of the study. Likewise, parts of the paper require clarifications and revision, as explained below.

98: Would be helpful to include a map in the supplementary to show the locations of the lakes. Not every reader is familiar with the geography of that region.

105: Here and elsewhere: units should be written in the more common format such as "h-1", etc.

107: Here and elsewhere: in-text citation format "refs. 8 and 10" etc. is not consistent with the others.

268: The discussion of δC , I think, is the weaker part of the paper. The authors used mixed assemblages for the experiments, which would have contained both methane producers and methane oxidizers. Their data suggested the presence of different types of producers using different pathways, of which the δC signatures were unknown. Their data also suggested that methane oxidation took place at variable rates in the incubation, which would also affect δC but to unknown extents. Without properly delineating the isotopic shifts from production vs. oxidation, it is rather unhelpful to speculate on the methane production source based on the observed δC . I think the experimental manipulation and the omic analysis together are already quite convincing in illustrating the organisms and pathways that underpin OWMP, even without the lengthy discussion of the δC data.

336: I think "lack of a δC isotopic signal" is a wrong expression. Fig. 3 shows that δC did not differ between treatments for L7 and LG6. One may have different interpretations of such observations, but it is certainly not a 'lack' of δC signal.

356: This paragraph seems overly speculative to me. The discovery of DPOR and COR is interesting and it is consistent with their observations (as well as others') that cyanobacteria are closely associated with OWMP. However, additional speculation about BES as electron donor or light disrupting photosynthesis is unnecessary and distracting.

367: Why another "Discussion" subheading? (see p. 5 "Results and Discussion") Based on my reading of the text, I think it is more appropriate to organize the text into separate "Results" and "Discussion" sections.

367: Further about the "Discussion" section:

As I stated earlier, the strength of the paper is the use of experiments and omic analysis to study the organisms and pathways behind the OWMP phenomenon. While the data are quite solid in that regard, the interpretation of δC is rather inadequate. There are other weaknesses I can identify in the paper:

(1) The lack of discussion/ explanation of the highly variable results between lakes. I appreciate the fact that even lakes close to each other geographically can be very different ecologically, but in this case there are no ancillary environmental data that will help readers understand why the results are so different between the lakes even though they are all located within the same vicinity, and how these results may be comparable (or not) to other lakes.

(2) The lack of discussion/ explanation of the very different results between "replicates" shown in Fig. 1 (notwithstanding the fact that they were not true replicates as indicated in Table S5). For example, according to Table S5, LG1, LG2 and LG3 should give similar results, but that was not the case. I also find it difficult to read Fig. 1 without referring to Table S5 for the different labels.

464: Wrong value and unit for light? If this is supposed to be the total photons for the entire experimental duration (11 hours), then I calculated a light level of only $12.6 \mu\text{mol m}^{-2} \text{s}^{-1}$, which is hardly any light at all!

475: Please include actual dissolved oxygen data, perhaps in the supplementary. Anaerobic methanogenesis could kick in even before the bulk-water D.O. reached zero. I think the authors need to show the data to convince readers that anaerobic methanogenesis was unlikely to occur in their experiments.

Fig.S1: A rather confusing graph: The y-axis says "change in methane concentration (%)" which is inconsistent with the figure caption saying "significant differences in CH₄ concentrations". At any rate, it is very misleading to plot percentage change (if that was indeed the case)--- Percentage change from what time point? Time zero, or the immediate time point before? Also, it is more meaningful to know the actual amount of CH₄ produced rather than percentage change as an indication of the actual methane production. It is also not appropriate to test for significant differences between percentages with ANOVA and Tukey HSD tests. Lastly, all the lines start off at zero, but it would be meaningless to show any 'percentage change' from a zero value.

Overall, the authors should refocus the discussion on the key findings, revise the weaker parts of the discussion and improve the overall organization and presentation.

Reviewer #2 (Remarks to the Author):

Perez-Coronel and Beman investigated oxic ('paradoxical') methane production in high-elevation lakes of the Yosemite National Park. For this, they used a combination of incubation experiments in conjunction with 16S rRNA gene and metagenomic sequencing. The authors conclude that multiple mechanisms, including two potentially new ones, co-occur in the investigated lakes and that the two novel mechanisms may be broadly distributed in aquatic ecosystems.

Oxic methane production is a topic of high relevance and very timely. I find it commendable that the authors chose an interdisciplinary approach to tackle this important research question, combining incubation experiments, analytical, and molecular techniques. This study could contribute to our knowledge on aerobic methane production in lakes, particularly regarding the link to primary production.

However, in its current state the manuscript has - in my opinion - several significant flaws, including an essentially non-existent results section, poorly described experimental methods chapter, and a very repetitive and convoluted discussion (there is

Results & Discussion, followed by a separate Discussion chapter). I had a hard time following the authors' argumentation and I have reservations regarding their main conclusions (see below). While I value the authors' effort to disentangle the mechanisms behind lacustrine methane production, in my opinion, there is too much ambiguity regarding their data interpretation to support the main claims made in the title and abstract regarding (i) novel methane-production mechanisms and (ii) broad distribution across aquatic ecosystems.

Regarding (i): all mechanisms that are discussed to possibly play a role in methane production in the investigated lakes (methanogenesis, MPn, DMSP and photosynthesis-linked methane production) have been described before and thus are not novel. The authors presumably refer to the two 'novel' pathways that they speculate may underlie the photosynthesis-linked mechanism but that these pathways can produce methane at all is not shown and the authors do not present evidence for the involvement of these 'novel' pathways in their investigated lakes. I list more specific comments below.

Regarding (ii), this conclusion seems to be based on the 'wide distribution' of the Comamonadaceae family in aquatic habitats. However, the authors do not present evidence that this family is indeed involved in methane production; instead they make this assumption based on the abundance of Comamonadaceae 16S rRNA gene in their treatments. However, this is misleading – as Table 1 shows, out of their 19 experiments, Comamonadaceae 16S genes or transcripts in fact decreased over time in 5 experiments and only increased in 1 experiment. It should be pointed out that this one experiment (UC3) is also the sole one incubation in which methane production followed a 'non-linear pattern'. Hence, this experiment is not representative of a universal, 'broadly distributed' mechanism and such generalization is out of place.

I acknowledge that the authors collected a large dataset from 19 sampling campaigns but their presentation of the results leaves many important issues open. Probably the most important one is reproducibility. The authors mention that they performed their experiments in triplicates. Yet, they do not show their data with standard deviations/errors and they do not comment on the differences among the respective triplicates for the individual treatments. As the conclusions of this manuscript are largely based on comparisons between sites and treatments, measured ranges of rates must be reported instead of a single rate.

Comments regarding the individual mechanisms

Methanogenesis:

The development of rates over time in BES-amended incubations is missing. Any comparisons are not conclusive as long as there is no mention of the biological variability in the experimental triplicates. Sometimes, the claims in the text contradict the data shown in Figures (eg. In Figure S2, BES had a significantly negative effect in 3 incubations and positive in 2 vs. Line 146 says decrease in 4 and increase in 4).

MPn:

Even though I agree with the authors that this mechanism is likely active in situ (line 381) their argument is based on indirect evidence only as no link between MPn and methane production is shown here. The authors do not report neither MPn, nor in situ phosphate concentrations, even though the latter are known to be the main regulating factor for this process. Regarding their molecular data, if the y axis is correct, we are looking at the upper range between 0.0001 and 0.001% relative abundance. This is a very low number and I am wondering how many total reads were retrieved for the different genes at all? I doubt it is possible to draw conclusions based on such low read numbers. Similarly, an average 2-fold 'up- and down-regulation' of these genes is unlikely to be relevant.

DMSP:

The authors mention that they found genes involved in the demethylation of DMSP

(dmdA-D; Line 199). However, there is no evidence to suggest that these genes are indeed implicated in methane production; demethylation of DMSP forms methylmercaptopropionate and eventually methanethiol, not methane. Additionally, the authors show no data regarding e.g. the presence of DMSP in the investigated lakes. Hence, I do not think that the authors can imply that DMSP may be a methane precursor in these lakes.

Photosynthesis + Novel mechanisms

To me, a causal relationship between photosynthesis/primary production and methane production is not clearly established. The discussion strongly focuses on the 'high-light' and BES experiments, which for one lake showed very high methane production rates. I must say I do not understand why BES and high light treatments are being discussed together, because as far as I'm aware, BES is a structural analogue of coenzyme M and has no connection to photosynthesis. The authors' suggestion that it could 'act as a source of electrons similar to thiosulfate' (for DPOR and COR? Unclear; line 361) is to me an unfounded speculation. Given the counterintuitive effect of BES on methane production the authors need to show convincing evidence that the stimulating effect of BES is reproducible and not a result of variability among replicates. I struggle to see how BES and high light could elicit the same effect in the incubations, apart from maybe inducing stress. Indeed, the authors suggest that one of their two 'novel' mechanisms might be stress-induced methane formation from methoxy groups of pigment precursors. I am not aware that such process has been shown to exist in bacteria and lead to methane formation at such high rates. Therefore, the authors would need to support their claim by additional data or literature evidence.

The other 'novel' mechanism implicates the enzymes DPOR and COR in methane production and the authors propose that these enzymes may reduce CO₂ to methane (line 359). Apart from the fact that such reaction has never been shown for these enzymes (to my knowledge), this speculation is incompatible with the authors' own delta 13C results, which suggest a much lighter source of methane.

Overall, the presented support for any of these novel photosynthesis-linked pathways is only indirect; hence, I would find it more appropriate to accordingly down-tone the conclusions regarding the novel pathways.

Reviewer #3 (Remarks to the Author):

The manuscript by Perez-Coronel & Beman presents a series of experiments to examine methane production in oxic surface lake water. The manuscript clearly lays out 3 possible sources for the methane (standard anaerobic methanogenesis perhaps in microenvironments, methylphosphonate catabolism, and photosynthesis-associated) and systematically assesses the potential contribution from each source. The major new claim of this paper is the proposal that photosynthetic bacteria (including cyanobacteria and aerobic anoxygenic phototrophs) could be producing methane via two enzymes, COR and DPOR. If this mechanism is borne out, then this paper will have a huge impact on the field.

The evidence presented for this mechanism is circumstantial: the two proposed enzymes are identified by transcriptomics in a single experiment ("UC4"). Because of this, any claims by the authors to have identified and demonstrated a new pathway for methane production should be tempered appropriately. It remains a proposal, not a demonstration. I believe it is intriguing and perhaps provocative enough to support publication. The authors combine several types of data/analysis to support their conclusion that a 3rd pathway is occurring (in addition to standard methanogenesis and methylphosphonate degradation). But the specific identification of the enzymes themselves (or an alternative suggestion, methoxyl groups from pigment biosynthesis intermediates) as the source, is less convincing. It may well be correct, but based on the data in this paper, should be carefully worded. (e.g. line 368, 431)

Several points were unclear:

1. The methods state that experimental incubations were done in triplicate, but there are no error bars shown on the figures. Error bars & clear replication would make this more convincing.

2. The transcriptomics and metagenomics analyses are not especially clear. (a) First, the authors should take care in reporting relative abundance values and their interpretation (e.g. lines 303-310, 324-331, 332-335, 401-402). The values themselves have little meaning without more context (e.g. comparison to other genes, normalization to housekeeping genes). Also, relative abundance can change due to either a change in the numerator or a change in the denominator. For example, the relative abundance of DPOR transcripts may appear higher if other transcripts in the pool are changing significantly – it does not require “upregulation” or “downregulation” of the targets themselves (fig 4 legend and text). A better approach might be to normalize these transcript abundances to housekeeping genes, and perhaps to show another functional gene for comparison.

(b) I did not follow the logic and methods for the read mapping to MAGs. Were the MAGs from this study or from a different study (GEM? Line 420, 561-2)? The transcript mapping (fig 5c) would be more informative if DNA reads were mapped to the MAG, and RNA normalized to DNA mapping. For example, it is possible that some sequences could recruit higher numbers of reads due to shared sequence identity (potentially across multiple taxa). The authors should be careful not to refer to the abundance of genes from Limnohabitans MAG 5 (lines 321-345) – there is not enough evidence to say that this strain exists in these incubations (or anywhere, since MAGs are composites). Rather, you find sequence with high similarity to this MAG.

(c) Were DPOR and COR the only two transcripts with differential abundance (line 296-298, 553-557)? Unclear from the text. Also unclear how differential abundance was determined (were there replicate RNA datasets sequenced?)

(d) unclear how annotations were assigned (line 546-8). By the text annotation of the database match? By Pfam/KEGG/etc?

3. Figures: explain abbreviations (e.g. fig 2 what are the treatments?). There is no need to color the positive/negative trends because their values show the directionality (could emphasize the zero line).

4. Sections are labeled “Results and Discussion” and “Discussion”. Maybe “Conclusions”? or “Results” & “Discussion”?

5. Title & abstract: should temper the claims. Not clear if “novel methane production mechanisms” is referring to the proposed DPOR/COR, but again it is a proposed mechanism. And “broadly distributed across aquatic ecosystems” – but really the evidence is from one lake/incubation. Abstract “identified” (line 24), “particularly active” (line 27), “identifying” (line 30) should be reworded.

Perez-Coronel and Beman - Response to Reviewers

We thank the reviewers for their very helpful comments on our original manuscript. We have addressed all of their comments in our revision, which we think has resulted in an improved manuscript. We have highlighted these changes in blue in the revised manuscript (although deleted text is not shown). Our specific responses to all reviewer comments are shown in blue below.

Reviewer #1 (Remarks to the Author):

The authors used a combination of incubation experiments and 'omic' analysis to study the phenomenon of oxic-water methane production (OWMP). They concluded that the production was functionally linked to bacterial chlorophyll and photosynthesis, and taxonomically to cyanobacteria and proteobacteria.

OWMP contradicts the prevailing paradigm of biological methanogenesis being a strictly anaerobic process, and it may explain the widely reported 'methane paradox' in aquatic systems. Because the idea goes against the 'text book' knowledge about methane cycle, it has been hotly contested--- not unusual for any ground-breaking idea in science. Nevertheless, the argument for OWMP will only strengthen when researchers continue to identify the organisms and biochemical mechanisms that underpin the phenomenon. In this regard, this paper offers some important and new findings. The experiments appear to be well executed. The use of omic techniques produces useful insights into the responsible organisms.

We thank the reviewer for these positive comments.

Overall, it is a welcomed contribution to OWMP research, but I find the paper a bit frustrating to read at times when the key messages are buried in the rather tedious description of the different observations in the different lakes and the different treatments. If the authors can improve the organization and presentation of the key findings, it would help readers to grasp the importance of the study. Likewise, parts of the paper require clarifications and revision, as explained below.

We apologize for the lack of organization and clarity in presenting some of our results. One of the challenges is that we conducted multiple experiments with multiple treatments, and all with multiple sampling time points. This permits a number of different ways of presenting the data and possible statistical comparisons. Our original approach was to describe patterns in the controls only, then mention nonlinear patterns over time, and then discuss the different treatments in the context of different processes that might be producing methane. In general, we tried to highlight specific examples by noting experiments and treatments in the text, often in parentheses. Obviously, this was not an effective approach. In our revised manuscript, we first describe the overall patterns across all experiment. We include both controls and experimental treatments in a revised Figure 1, and describe how we expanded our experiments in time and number of treatments. Then we summarize cases where we saw nonlinear patterns, followed by no significant trends, and finally experiments with decreases in methane.

Throughout the manuscript, we have worked to improve the clarity and have removed instances where discussing particular lakes or treatments is not needed. We also feel that removal of some less important text (as suggested by various reviewers) helps with the organization. In particular, we removed the text about DMSP (see below in response to Reviewer 2), and have removed some of the text regarding the effects of BES in the UC4 experiment (see below).

98: Would be helpful to include a map in the supplementary to show the locations of the lakes. Not every reader is familiar with the geography of that region.

We now include a map of the study locations in the supplemental material.

105: Here and elsewhere: units should be written in the more common format such as “h⁻¹”, etc.

We apologize for this oversight and have fixed units throughout the paper.

107: Here and elsewhere: in-text citation format “refs. 8 and 10” etc. is not consistent with the others.

Based on the guide to authors, references are not displayed in the typical superscript format when this may produce confusion, especially after numbers/symbols/equations. We agree that this is unnecessary in this case (this is a holdover from an earlier draft) and have fixed it.

268: The discussion of δC , I think, is the weaker part of the paper. The authors used mixed assemblages for the experiments, which would have contained both methane producers and methane oxidizers. Their data suggested the presence of different types of producers using different pathways, of which the δC signatures were unknown. Their data also suggested that methane oxidation took place at variable rates in the incubation, which would also affect δC but to unknown extents. Without properly delineating the isotopic shifts from production vs. oxidation, it is rather unhelpful to speculate on the methane production source based on the observed δC . I think the experimental manipulation and the omic analysis together are already quite convincing in illustrating the organisms and pathways that underpin OWMP, even without the lengthy discussion of the δC data.

We agree that our interpretation of these results was too simplistic and have modified this section. We added discussion of methane oxidation, which we had originally included in the manuscript but elected to remove prior to submission. Our original reason for removing this was that methane oxidation is likely low given the lack of change in concentrations and isotopic signatures in the controls, and the fact that high light can inhibit methane oxidation. We now state all of this in the manuscript, and also note that appreciable methane oxidation implies an even lighter source of CH₄. So, we now simply note that the source must be less than -55 per mil (and we could also include estimates of the values assuming different methane oxidation rates if desired). Finally, for this revision, we also analyzed samples from the UC3 experiment, which showed a similar response in the BES treatment (and there was no high-light treatment).

However, taking a step back, the data speak for themselves because there are clear differences between the treatments and the control in the UC4 (and UC3) experiment, and these differences clearly contrast with the other experiments. Again, this is an advantage of using experimental manipulations rather than only using observational data. Overall, the patterns in our data are quite distinct, and indicate that the CH₄ being produced is isotopically depleted. This can be used to infer sources—or at least exclude some—which we do in the text. However, we did delete a portion of the text comparing this to other work with phytoplankton and other lakes. We now simply state that this implies another source. For all of these reasons, we think that these data are worth including in revised form.

336: I think “lack of a δC isotopic signal” is a wrong expression. Fig. 3 shows that δC did not differ between treatments for L7 and LG6. One may have different interpretations of such observations, but it is certainly not a ‘lack’ of δC signal.

We apologize for this phrasing. As noted above, we are just highlighting the strong signal in the UC4 experiment, but the reviewer is correct that this doesn’t mean a ‘lack of signal’ in the other experiments. We rephrased this to “These patterns are consistent with the changes in $\delta^{13}\text{CH}_4$ that were observed in the UC4 experimental treatments, but not in the LG6 and L7 experiments”

356: This paragraph seems overly speculative to me. The discovery of DPOR and COR is interesting and it is consistent with their observations (as well as others’) that cyanobacteria are closely associated with OWMP. However, additional speculation about BES as electron donor or light disrupting photosynthesis is unnecessary and distracting.

We agree with the reviewer and have deleted the text focused on BES and light from this paragraph.

367: Why another “Discussion” subheading? (see p. 5 “Results and Discussion”) Based on my reading of the text, I think it is more appropriate to organize the text into separate “Results” and “Discussion” sections.

This was a careless mistake that was noted by multiple reviewers, and we apologize for this! Following this suggestion, we now have separate Results and Discussion sections.

367: Further about the “Discussion” section:

As I stated earlier, the strength of the paper is the use of experiments and omic analysis to study the organisms and pathways behind the OWMP phenomenon. While the data are quite solid in that regard, the interpretation of δC is rather inadequate. There are other weaknesses I can identify in the paper:

(1) The lack of discussion/ explanation of the highly variable results between lakes. I appreciate the fact that even lakes close to each other geographically can be very different ecologically, but in this case, there are no ancillary environmental data that will help readers understand why the results are so different between the lakes even though they are all located within the same vicinity, and how these results may be comparable (or not) to other lakes.

We agree that additional environmental information is useful and that this was an oversight in our initial submission. We have included some of these data in a supplemental table. A key aspect of this is that these lakes are spread along an elevation gradient, and so differ in various ways. We now note this in the Introduction and refer to the new online table.

We did mention in our previous submission several underlying reasons for variation. In particular, we discussed variation in methane oxidation in the first paragraph of the Discussion, and competition for different forms of P in the second paragraph of the Discussion. We mentioned variations in photosynthesis in the Results as well as in the Discussion, and specifically that aerobic anoxygenic photosynthesis (which we still know comparatively little about) is likely variable. In addition, nearly every study that has examined paradoxical methane production has shown that it is quite variable, and so our study is not alone in this regard. To emphasize this point, we added a sentence to the Introduction that specifically states that paradoxical CH₄ production can vary by orders of magnitude over the course of days.

All of this said, we agree that there are some variations between lakes and over time in lakes (see below) that need to be noted. Please see our response directly below about this.

(2) The lack of discussion/ explanation of the very different results between “replicates” shown in Fig. 1 (notwithstanding the fact that they were not true replicates as indicated in Table S5). For example, according to Table S5, LG1, LG2 and LG3 should give similar results, but that was not the case. I also find it difficult to read Fig. 1 without referring to Table S5 for the different labels.

From these and other comments, it is clear that we need to describe our experiments more clearly. First, a key point is that we did have true replicates in all of the experiments—as well as in all treatments and time points in all experiments. We have included additional detail in the methods to clarify that sampling was sacrificial and in triplicate at each time point. We also agree that Figure 1 didn't fully convey this replication, especially with a lack of standard errors, which we have now added. Our intention was actually to accomplish what reviewer 1 highlights above, which is an overview of the experiments without going into too much detail that might lead to confusion.

Second, the LG1-LG3 experiments took place in different months, and, as we note above, paradoxical methane production is variable over time. But in the specific case of LG2, two outliers led to a lack of significance that would otherwise make all three experiments similar. We now specifically note this in the text, and feel it is not prudent to exclude these outliers.

In addition to adding the standard errors to Figure 1, we expanded this figure to include experimental treatments rather than just the controls. We also provided additional context by showing the years in which different experiments took place. We have kept the Table with experimental information in the supplemental, but would certainly consider moving it to the main text if that would provide additional clarity.

Finally, to address the comments above and here, we have added some additional text about the characteristics of our sampled lakes that may lead to variations in CH₄ production in space and time. (But again, we note that this has been observed in nearly every study to date.) We added additional text about variations in different forms of P, as well as different sources of P and N, to the second paragraph of the Discussion. (Related to this, we analyzed the new paradoxical CH₄ production mechanism associated with methylamine metabolism and included some information on this in the Results and Discussion.) Our point here is that all of this vary in space and time, and so can lead to variation in use of different substrates, and therefore CH₄ production. We also added additional text to the third paragraph of the Discussion focused on factors regulating AAP and how they may result in variation. Finally, an overarching aspect of our study is that if there are multiple CH₄ production mechanisms active, as well as possible variations in methane oxidation, then it is no surprise that paradoxical methane production is quite variable. We now directly state this in the text.

464: Wrong value and unit for light? If this is supposed to be the total photons for the entire experimental duration (11 hours), then I calculated a light level of only 12.6 $\mu\text{mol m}^{-2} \text{s}^{-1}$, which is hardly any light at all!

These were the wrong units and have been fixed accordingly.

475: Please include actual dissolved oxygen data, perhaps in the supplementary. Anaerobic methanogenesis could kick in even before the bulk-water D.O. reached zero. I think the authors need to show the data to convince readers that anaerobic methanogenesis was unlikely to occur in their experiments.

We apologize for not including this before, but these are high elevation lakes with low levels of productivity and carbon/nutrient input, and so incubations will almost certainly stay well-oxygenated. However, we now include the DO data in a Table in the supplement, and mention in the methods that all were well-oxygenated.

Fig.S1: A rather confusing graph: The y-axis says "change in methane concentration (%)" which is inconsistent with the figure caption saying "significant differences in CH₄ concentrations". At any rate, it is very misleading to plot percentage change (if that was indeed the case) --- Percentage change from what time point? Time zero, or the immediate time point before? Also, it is more meaningful to know the actual amount of CH₄ produced rather than percentage change as an indication of the actual methane production. It is also not appropriate to test for significant differences between percentages with ANOVA and Tukey HSD tests. Lastly, all the lines start off at zero, but it would be meaningless to show any 'percentage change' from a zero value.

We apologize for the confusion with this graph. Our intention was to normalize different experiments in order to present multiple experiments on the same graph. We did conduct our statistics on the actual data, not percentages, but obviously all of this was unclear. To resolve

these issues, we scrapped this figure and instead include data from the different treatments in Figure 1.

Overall, the authors should refocus the discussion on the key findings, revise the weaker parts of the discussion and improve the overall organization and presentation.

Reviewer #2 (Remarks to the Author):

Perez-Coronel and Beman investigated oxic ('paradoxical') methane production in high-elevation lakes of the Yosemite National Park. For this, they used a combination of incubation experiments in conjunction with 16S rRNA gene and metagenomic sequencing. The authors conclude that multiple mechanisms, including two potentially new ones, co-occur in the investigated lakes and that the two novel mechanisms may be broadly distributed in aquatic ecosystems.

Oxic methane production is a topic of high relevance and very timely. I find it commendable that the authors chose an interdisciplinary approach to tackle this important research question, combining incubation experiments, analytical, and molecular techniques. This study could contribute to our knowledge on aerobic methane production in lakes, particularly regarding the link to primary production.

We thank the reviewer for these positive comments.

However, in its current state the manuscript has - in my opinion - several significant flaws, including an essentially non-existent results section, poorly described experimental methods chapter, and a very repetitive and convoluted discussion (there is Results & Discussion, followed by a separate Discussion chapter). I had a hard time following the authors' argumentation and I have reservations regarding their main conclusions (see below). While I value the authors' effort to disentangle the mechanisms behind lacustrine methane production, in my opinion, there is too much ambiguity regarding their data interpretation to support the main claims made in the title and abstract regarding (i) novel methane-production mechanisms and (ii) broad distribution across aquatic ecosystems.

Regarding (i): all mechanisms that are discussed to possibly play a role in methane production in the investigated lakes (methanogenesis, MPn, DMSP and photosynthesis-linked methane production) have been described before and thus are not novel. The authors presumably refer to the two 'novel' pathways that they speculate may underlie the photosynthesis-linked mechanism but that these pathways can produce methane at all is not shown and the authors do not present evidence for the involvement of these 'novel' pathways in their investigated lakes. I list more specific comments below.

With regard to (i) novel production mechanisms, we use 'novel' in the title to refer to all oxic/paradoxical sources of methane. As highlighted by Reviewers 1 and 2 above, and the discussion surrounding the Günthel et al. (2019) study in *Nature Communications*, this remains

timely but still hotly contested. However, we agree that 'novel' is unclear in the title, and so have changed this to 'paradoxical.' We do present clear evidence that methane is produced paradoxically, which supports point (i).

With regard to specific mechanisms and pathways, it is important to note that the vast majority of information in the literature is based on observations alone, without insight into the possible mechanisms and pathways that produce methane paradoxically. For example, Bogard et al. in *Nature Communications* and Günthel et al. both show production of CH₄, with some additional evidence in the form of isotopic data and correlations with environmental data. Likewise, Bizic et al. demonstrate cyanobacterial production of CH₄, but to be entirely accurate, they do not show exactly how CH₄ is produced by Cyanobacteria. They infer from their observations over light-dark cycles that CH₄ is likely produced during photosynthesis itself, and they also rule out N-fixation—but exactly which genes/enzymes are involved remains unclear. In this context, we disagree that we haven't provided evidence for different pathways. Our experimental data, treatment effects, multi-omic data, and isotopic data all provide evidence for these pathways.

Our data point strongly towards CH₄ production by Comamonadaceae during aerobic anoxygenic photosynthesis. Although this is obviously a form of photosynthesis, it is clearly distinct from oxygenic photosynthesis. Additional evidence would come in the form of pure culture work analogous to that of Bizic et al., and is a logical next step. But first one has to know what to look for, and our study provides this key information. We analyze pathways and organisms in a variety of experimental treatments, which provides both new insight into previous observations, as well as evidence for potentially new mechanisms.

Regarding (ii), this conclusion seems to be based on the 'wide distribution' of the Comamonadaceae family in aquatic habitats. However, the authors do not present evidence that this family is indeed involved in methane production; instead they make this assumption based on the abundance of Comamonadaceae 16S rRNA gene in their treatments. However, this is misleading – as Table 1 shows, out of their 19 experiments, Comamonadaceae 16S genes or transcripts in fact decreased over time in 5 experiments and only increased in 1 experiment. It should be pointed out that this one experiment (UC3) is also the sole one incubation in which methane production followed a 'non-linear pattern'. Hence, this experiment is not representative of a universal, 'broadly distributed' mechanism and such generalization is out of place.

The reviewer raises some important points here, but there are multiple inaccuracies. First, we do not rely solely on the abundance of 16S genes in our experiments, we also present metagenomic and, importantly, metatranscriptomic data. As noted in multiple locations in the manuscript, the genes and transcripts recovered in the omic data are similar and in some cases identical to database sequences from other locations.

In addition, one would not necessarily expect an increase in abundance linked to CH₄ production, as this assumes that CH₄ production is directly linked to growth. While this is obviously true for conventional methanogenesis, CH₄ production via paradoxical mechanisms

may not be a core aspect of energy generation and growth. Even for oxygenic photosynthesis, CH₄ is produced in an ancillary way (e.g., in contrast with the production of oxygen itself). (Note that this is also important when considering the isotopic data; see below). More broadly, the (ii) wide distribution across ecosystems relies on multiple sources of data—and we provide additional evidence for this our revision.

In particular, we previously stated the similarity of metatranscriptome and metagenome data to database sequences, but we have included additional information through comparison with freshwater metagenomes present in the JGI IMG/M database. The take home point here is that similar and even identical sequences to those recovered from Yosemite are found in freshwater ecosystems worldwide, and this is true for multiple genes. In fact, our *Limnohabitans* MAG 5 shares >95% average nucleotide identity to several MAGs from lakes in Canada, suggesting wide distribution. We further demonstrate this, and include a few sentences in the paper and an online Table.

We also examined available 16S databases. We found that our *Limnohabitans* MAG 5, MAGs from Canadian Lakes sharing >95% ANI to MAG 5, and our dominant Comamonadaceae ASVs all fell within an operational taxonomic unit (OTU) that was previously identified in the “Global Prokaryotic Census” of Louca et al. We found that this OTU is distributed across several thousand freshwater samples worldwide. For brevity, we elected not to include these data in the revision, but would certainly include them in the supplemental if it helps further support our point.

Finally, we did make several other modifications to the title that align more closely with what we measured and observed in our study.

I acknowledge that the authors collected a large dataset from 19 sampling campaigns but their presentation of the results leaves many important issues open. Probably the most important one is reproducibility. The authors mention that they performed their experiments in triplicates. Yet, they do not show their data with standard deviations/errors and they do not comment on the differences among the respective triplicates for the individual treatments. As the conclusions of this manuscript are largely based on comparisons between sites and treatments, measured ranges of rates must be reported instead of a single rate.

Please see our responses to Reviewer 1 above about replication. We agree that this was unclear, and have clarified it in the methods. We also apologize for the lack of error bars, and we now report standard errors. We previously used Spearman correlation to test for significance, but modified this to Pearson, as most experiments were significant for both. This provides error estimates for the slopes (rates) of CH₄ vs. time, which are included in Figure 1.

Comments regarding the individual mechanisms

Methanogenesis:

The development of rates over time in BES-amended incubations is missing. Any comparisons are not conclusive as long as there is no mention of the biological variability in the experimental triplicates. Sometimes, the claims in the text contradict the data shown in Figures (eg. In Figure S2, BES had a significantly negative effect in 3 incubations and positive in 2 vs. Line 146 says decrease in 4 and increase in 4).

We agree that we did not fully describe these data in our previous submission and have fixed this. We now summarize the patterns over time in this section—which we agree is important to do, as several BES treatments show increases in CH₄ over time. These data are also directly reported in Figure 1 in the revised manuscript. As noted above, we also include error bars that show the variation in replicates. In addition, we modified Figure S2 to only show cases where treatment effects were significant, and the log response ratio of those effects. We previously included response ratios at particular timepoints for all the treatments, and then noted whether they were significant or not. As the reviewer noted, this is not what we discuss in the text, and so we apologize for confusion here.

MPn:

Even though I agree with the authors that this mechanism is likely active in situ (line 381) their argument is based on indirect evidence only as no link between MPn and methane production is shown here. The authors do not report neither MPn, nor in situ phosphate concentrations, even though the latter are known to be the main regulating factor for this process. Regarding their molecular data, if the y axis is correct, we are looking at the upper range between 0.0001 and 0.001% relative abundance. This is a very low number and I am wondering how many total reads were retrieved for the different genes at all? I doubt it is possible to draw conclusions based on such low read numbers. Similarly, an average 2-fold 'up- and down-regulation' of these genes is unlikely to be relevant.

These comments are not totally accurate. As the reviewer is probably aware, measuring MPn is very challenging, and so very few papers report MPn data of any kind. In previous methane paradox work, MPn was quantified indirectly by using NMR-based analysis of dissolved organic P (DOP) to quantify the relative proportion of DOP that is MPn. We are aware of limited measurements made in marine waters using this NMR-based approach (e.g., work of Dan Repeta and others), but none in freshwater. Wang et al. recently reported two attempted measurements of MPn via LC-MS in Yellowstone Lake, but MPn was below detection, and the specifics of the methods used to measure MPn were not reported. So our study is not alone in lacking measurements of MPn, as they are rare at best.

However, we did measure phosphate concentrations, as we previously stated in our manuscript on lines 223-226: "inorganic phosphate concentrations in the lakes sampled here are consistently near the limits of detection (100 nM) by colorimetric techniques, with the vast majority of measurements <200 nM." In addition to this statement, we now include these data in an online table to make this clear.

Regarding the omics data, these datasets capture multiple organisms and multiple genes/transcripts within these organisms. Many of these organisms have different roles, and contain genes for many different functions. Acquiring P is just one of these functions, and only a subset of organisms can use MPn. So, we would not expect high gene abundances for something that only some organisms are using to acquire a single resource. However, we agree that the low percentages are confusing, and Reviewer 3 also suggested that we normalize our data to housekeeping genes. We normalized to *recA* genes, and these data are reported in Figures 2 and 5. However, it is important to note that the *recA* gene numbers generally scale with overall read numbers, so these data are generally similar. In the MPn section, we do not emphasize any comparatively minor quantitative differences between samples; our point is that *phnJ* is found in all of them.

DMSP:

The authors mention that they found genes involved in the demethylation of DMSP (*dmdA-D*; Line 199). However, there is no evidence to suggest that these genes are indeed implicated in methane production; demethylation of DMSP forms methylmercaptopropionate and eventually methanethiol, not methane. Additionally, the authors show no data regarding e.g. the presence of DMSP in the investigated lakes. Hence, I do not think that the authors can imply that DMSP may be a methane precursor in these lakes.

We removed the discussion of DMSP from the manuscript. Several papers in the literature and several talks at recent meetings have raised this as a possibility, so it will be interesting to see whether this ends up being significant in freshwater lakes.

Photosynthesis + Novel mechanisms

To me, a causal relationship between photosynthesis/primary production and methane production is not clearly established. The discussion strongly focuses on the 'high-light' and BES experiments, which for one lake showed very high methane production rates. I must say I do not understand why BES and high light treatments are being discussed together, because as far as I'm aware, BES is a structural analogue of coenzyme M and has no connection to photosynthesis. The authors' suggestion that it could 'act as a source of electrons similar to thiosulfate' (for DPOR and COR? Unclear; line 361) is to me an unfounded speculation. Given the counterintuitive effect of BES on methane production the authors need to show convincing evidence that the stimulating effect of BES is reproducible and not a result of variability among replicates. I struggle to see how BES and high light could elicit the same effect in the incubations, apart from maybe inducing stress. Indeed, the authors suggest that one of their two 'novel' mechanisms might be stress-induced methane formation from methoxy groups of pigment precursors. I am not aware that such process has been shown to exist in bacteria and lead to methane formation at such high rates. Therefore, the authors would need to support their claim by additional data or literature evidence.

The other 'novel' mechanism implicates the enzymes DPOR and COR in methane production and the authors propose that these enzymes may reduce CO₂ to methane (line 359). Apart from the fact that such reaction has never been shown for these enzymes (to my knowledge),

this speculation is incompatible with the authors' own delta 13C results, which suggest a much lighter source of methane.

Overall, the presented support for any of these novel photosynthesis-linked pathways is only indirect; hence, I would find it more appropriate to accordingly down-tone the conclusions regarding the novel pathways.

Given these comments and those of Reviewer 3, we toned down our conclusions. However, there are several aspects of this that are important to understand. First—and as we stated above—the published evidence for photosynthetic production of CH₄ in lakes and by cyanobacterial cultures is also 'indirect' in that it is based almost entirely on observational data. We present omic data that can provide more insight. Second, we observed the BES effect in multiple experiments, although the strongest effect is in UC4 and we have the most information from this experiment. For this revision, we also analyzed isotopic values from the UC3 experiment, which were highly similar to UC4. Third, the reason these incubations are discussed with the high-light treatment is that they produced similar patterns. Finally, production of CH₄ from CO₂ can only represent a relatively small percentage of overall CO₂ fixation—otherwise there would be extremely large rates of CH₄ production—and so represents a small 'leak' in the process. This would favor the lighter isotope, leading to isotopic fractionation. The isotopic composition of any produced CH₄ will not directly equate to that of CO₂. Instead, it should be lighter—which is exactly what we observe.

In fact, a recent paper shows that production of CH₄ from CO₂ by nitrogenase is a highly fractionating process (by 43-67 per mil), similar to conventional methanogenesis. As we note in our manuscript, nitrogenases are now recognized to be promiscuous enzymes that can produce CH₄ and other reduced compounds, and DPOR and COR fall within the family. We identified DPOR and COR based on omic data, and other explanations for our combined data are lacking. As we note above, here, below, and in the manuscript, our data therefore suggest another possible mechanism for paradoxical methane production.

Reviewer #3 (Remarks to the Author):

The manuscript by Perez-Coronel & Beman presents a series of experiments to examine methane production in oxic surface lake water. The manuscript clearly lays out 3 possible sources for the methane (standard anaerobic methanogenesis perhaps in microenvironments, methylphosphonate catabolism, and photosynthesis-associated) and systematically assesses the potential contribution from each source. The major new claim of this paper is the proposal that photosynthetic bacteria (including cyanobacteria and aerobic anoxygenic phototrophs) could be producing methane via two enzymes, COR and DPOR. If this mechanism is borne out, then this paper will have a huge impact on the field.

We thank the reviewer for these positive comments, and we agree with the point below that "It remains a proposal, not a demonstration. I believe it is intriguing and perhaps provocative enough to support publication."

The evidence presented for this mechanism is circumstantial: the two proposed enzymes are identified by transcriptomics in a single experiment (“UC4”). Because of this, any claims by the authors to have identified and demonstrated a new pathway for methane production should be tempered appropriately. It remains a proposal, not a demonstration. I believe it is intriguing and perhaps provocative enough to support publication. The authors combine several types of data/analysis to support their conclusion that a 3rd pathway is occurring (in addition to standard methanogenesis and methylphosphonate degradation). But the specific identification of the enzymes themselves (or an alternative suggestion, methoxyl groups from pigment biosynthesis intermediates) as the source, is less convincing. It may well be correct, but based on the data in this paper, should be carefully worded. (e.g. line 368, 431)

Several points were unclear:

1. The methods state that experimental incubations were done in triplicate, but there are no error bars shown on the figures. Error bars & clear replication would make this more convincing.

Please see our comments above in response to Reviewers 1 and 2 about this issue. In short, we calculated standard errors and have added these to Figure 1. We also more clearly describe the replication in the methods.

2. The transcriptomics and metagenomics analyses are not especially clear. (a) First, the authors should take care in reporting relative abundance values and their interpretation (e.g. lines 303-310, 324-331, 332-335, 401-402). The values themselves have little meaning without more context (e.g. comparison to other genes, normalization to housekeeping genes). Also, relative abundance can change due to either a change in the numerator or a change in the denominator. For example, the relative abundance of DPOR transcripts may appear higher if other transcripts in the pool are changing significantly – it does not require “upregulation” or “downregulation” of the targets themselves (fig 4 legend and text). A better approach might be to normalize these transcript abundances to housekeeping genes, and perhaps to show another functional gene for comparison.

We agree with this point and have normalized our data to the housekeeping gene *recA*. As we noted in our response to Reviewer 2 above, *recA* genes/transcripts generally scaled with the overall number of reads, and so our conclusions are essentially unchanged. But we agree that this is a better way to present the data in Figures 2 and 5. Regarding Figure 4, this presents the BES and high-light treatments in comparison to the control. We have clarified this in the figure legend and in the text. We also included *recA* on this figure for comparison. Finally, we changed any uses of ‘up- or downregulation’ to refer to increased or decreased expression.

(b) I did not follow the logic and methods for the read mapping to MAGs. Were the MAGs from this study or from a different study (GEM? Line 420, 561-2)? The transcript mapping (fig 5c) would be more informative if DNA reads were mapped to the MAG, and RNA normalized to DNA mapping. For example, it is possible that some sequences could recruit higher numbers of reads due to shared sequence identity (potentially across multiple taxa). The authors should be careful not to refer to the abundance of genes from Limnohabitans MAG 5 (lines 321-345) –

there is not enough evidence to say that this strain exists in these incubations (or anywhere, since MAGs are composites). Rather, you find sequence with high similarity to this MAG.

We have a sequencing grant from JGI and the MAGs are part of this project, which we clarified in the text. (We also have several other metagenomes in the process of sequencing, and waited some time to see if these could be included in this revision, but elected to move forward.)

We did use an approach that is similar to what the reviewer suggests, but did not include these data in the paper. First, it is important to note that we initially examined all reads, and noted the strong signal in DPOR and COR, particularly for the Proteobacteria. This led us to examine specific MAGs in greater detail. However, we also assembled metagenomes and mapped reads to the assemblies. We elected to not include these data in the revision, as this provides essentially the same conclusion because the signal is very strong. If the reviewer would like them to be included, we are happy to do so, but we felt that it is somewhat redundant. We include one approach that provides a broad view, and then focus in on Limnohabitans based on that overall view. Including an additional approach may be overkill, but we are happy to do so if requested.

Finally, we rephrased lines 321-345 to refer to sequences with high similarity to the MAG genes, rather than to the genes themselves.

(c) Were DPOR and COR the only two transcripts with differential abundance (line 296-298, 553-557)? Unclear from the text. Also unclear how differential abundance was determined (were there replicate RNA datasets sequenced?)

DPOR and COR showed the greatest changes in abundance, although there were other significant changes that were less pronounced. We now clarify this in the text. We also clarified the methods used to determine differential abundance. We did not sequence replicates, as this is atypical to do owing to the expense of these analyses. The IMG comparison tool computes z-scores across samples as described in this paper, which we now paraphrase and reference: <https://www.ncbi.nlm.nih.gov/pmc/articles/PMC2238950/>

(d) unclear how annotations were assigned (line 546-8). By the text annotation of the database match? By Pfam/KEGG/etc?

We clarified this in the methods to note that we queried reads using the BLASTX function in DIAMOND and then filtered the DIAMOND-generated output.

3. Figures: explain abbreviations (e.g. fig 2 what are the treatments?). There is no need to color the positive/negative trends because their values show the directionality (could emphasize the zero line).

We have made multiple changes to all of the figures and legends that should help clarify everything. In Figure 1, we now include different treatments and use color to identify these.

Lakes, years, and experiments are also shown. In Figure 2, we normalized the data and explain the abbreviations. In Figure 3, we added additional data. In Figure 4, we clarify that these data are shown relative to the control. And in Figure 5, we also normalized the data.

4. Sections are labeled “Results and Discussion” and “Discussion”. Maybe “Conclusions”? or “Results” & “Discussion”?

Again, we apologize for this mistake and now have separate “Results” and “Discussion” sections.

5. Title & abstract: should temper the claims. Not clear if “novel methane production mechanisms” is referring to the proposed DPOR/COR, but again it is a proposed mechanism. And “broadly distributed across aquatic ecosystems” – but really the evidence is from one lake/incubation. Abstract “identified” (line 24), “particularly active” (line 27), “identifying” (line 30) should be reworded.

We changed the wording in the Abstract for all cases noted by the reviewer. In general, we used ‘suggest’ rather than ‘identify’ in the Abstract. As noted above in response to Reviewer 2, we also changed ‘novel’ to ‘paradoxical’ in the title. Please see our response to Reviewer 2 about this, and regarding the broad distribution, which relies on multiple sources of data. Finally, we modified the title to temper our claims and more directly reflect what our data show.

Reviewer #1 (Remarks to the Author):

I commend the authors for their detailed answers to the previous questions/ comments, and careful and extensive revision of the manuscript. I find the revised manuscript much improved from the previous version; key messages are now presented clearly and easy to follow; figures and tables are also improved.

Overall this manuscript presents some new and interesting insights into the "paradoxical" oxic methane production in freshwaters. Granted, there are still many unknowns, but this study is a welcomed addition to this field of research.

The authors have addressed most of my previous comments satisfactorily, and I have only a couple of rather minor suggestions--- they are indeed only minor suggestions; the authors may or may not take them into consideration.

line 44: Change "indicates" to "may indicate". There are, frankly, still many who reject the notion that CH₄ can be produced under oxic conditions.

line 208: "We found that aat sequences from all but..." Do you mean all bacterial genera?

line 296: "enrich the remaining CH₄ pool" Do you mean enrich the isotopic composition of the remaining CH₄ pool?

line 312-387 and related Discussion: The results related to DPOR and COR are indeed very interesting and they support the idea of a photosynthesis-driven oxic CH₄ production pathway. However, based on your explanation, DPOR and COR actions do not directly produce CH₄ (nor based on my own understanding of the functions of DPOR nad COR). Likewise, it is still unclear to me how to explain the positive effect of BES on CH₄ production in this case. Are you suggesting that BES acts as electron donor in the reaction? On line 380 it is suggested that adding BES causes "stress" but this seems rather vague. Overall, I think you have very intriguing and worthy observations but there may be still missing pieces in the puzzle. Perhaps add a few words to line 468-472 to point readers to some future research questions.

Reviewer #2 (Remarks to the Author):

Review of Perez-Colonel and Beman (NCOMMS-21-09510A)

This is a revised version of a manuscript I reviewed earlier. I very much appreciate the effort the authors put into amending the manuscript according to mine and other reviewers' suggestions, and I believe these changes have now made the manuscript better focused and stronger.

1. The authors have addressed the concern about the 'novelty' of the methane-producing mechanisms by now replacing 'novel' with the term 'paradoxical', which I agree is better. They also adjusted the main claims in the abstract and throughout the text to better reflect their results and data.
2. The link between Comamonadaceae, their potential involvement in methane production, and their distribution in aquatic systems is spelled out more clearly now and easier to follow.
3. The reproducibility issue has been addressed by including standard errors into the graphs.
4. The distribution and prevalence of the genetic potential for MPn utilization (ie. the presence and expression of the phnJ gene) is described more clearly and is now discussed in the context of the ecological data.
5. My reservations about the 'novel' DPOR/COR-based mechanism have also been partially addressed. The authors toned down some of their more speculative

conclusions, and now more carefully outline their evidence regarding the potential involvement of these enzymes during photosynthesis-linked methane production. It seems that we agree that the link ultimately tying these enzymes to methane production is missing but the authors present enough data to allow for this interesting speculation.

My biggest outstanding comment is that the discussion of the multiple proposed pathways and the many lines of evidence for and against the various mechanisms remains rather convoluted. The authors seem to be aware of this problem and made some improvements during the revision; nonetheless, some conclusions still do not come across clearly, the biggest one for me perhaps – what is the main process/organisms contributing to aerobic methane production in this system? Is there even a main process or are all the different processes more or less equal? Based on the title 'methane production associated with photosynthesis and nutrient acquisition' I get the feeling it is rather the latter (possibly with the exception of methanogenesis) but more clarity regarding this would be needed.

Other comments:

Abstract, Line 29: evidence is not provided that Proteobacteria 'metabolized methylphosphonate' but that they encoded and expressed the genes to degrade methylphosphonate.

Line 46: Please check. It was my understanding that the 'methane paradox' pertains to methane supersaturation in oxic water not its production per se - albeit it might be used interchangeably now.

Line 73: 'similar' is misleading here. What specifically is the similarity referring to? It is not the enzymatic machinery, neither the regulation.

Line 108: CH₄ concentration measurements, presumably?

Line 120-121: sentence 'Given these consistencies...' is in my opinion redundant. It appears to suggest that due to the similarity in the magnitude of the rates, the rates appear broadly relevant, which does not make sense to me.

Line 122-125: It is unclear what the 'initial' and 'later' experiments refer to? In the figures and throughout the text the experiments are described under their acronyms (eg. LG1, UC) so it is not immediately obvious which ones are considered 'initial' and which ones 'later'. Is this distinction even important? Is it not enough to say some incubations were 24 hours and some up to 96 hours long?

Line 160: add 'methane' between 'showed' and 'consumption' to avoid ambiguity.

Line 171: 'Collectively, these data provide limited evidence...for water column-based methanogenesis as source of CH₄.' This sentence needs additional clarification. What type of methanogenesis do you mean, ie. which carbon compound do you presume would be the precursor of CH₄? Archaeal methanogens can produce CH₄ from CO₂, acetate or methylated compounds such as methyl-amines and methyl-sulfides. Please elaborate.

Line 178-179: 'other than conventional methanogenesis' is misleading. Classical euryarchaeal methanogens are also capable of using methylamines as methane precursors (ie. methylotrophic methanogens). The authors presumably want to highlight that methylamine should be demethylated via an aerobic - non-methanogenic - pathway. This should be clarified.

Line 186: what is meant with 'universally'?

Line 188: remove 'and active'. 'Active' would traditionally refer to enzymes and proteins rather than transcripts.

Line 215: '16S data' should read '16S rRNA gene data'.

Line 253-255: Sentence 'In addition, ...' should be removed from here. The sentence describes the recovery of aat genes but is placed in a paragraph that deals with methylphosphonate metabolism; this is confusing. Additionally, it is repetitive with the information that was provided in more detail in the paragraph on line 204ff.

Line 300: This paragraph somewhat confuses me. The authors discuss the measured methane $\delta^{13}\text{C}$ values and conclude that the data 'suggest another paradoxical CH₄ production mechanism'. First of all, it is unclear to me which other production mechanism this refers to. Is it the DPOR/COR pathway? In cyanobacteria or AANP? What would be the substrate for the DPOR/COR pathway, CO₂? Do the authors propose

this pathway would result in fractionation as high as methanogenesis?

Next, related to the methanogenesis pathway, as mentioned earlier, the authors seem to use this term interchangeably with methanogenesis from CO₂. However, there are three 'classical' methanogenic pathways, which use CO₂, acetate and methylated compounds, respectively. These pathways result in different isotope effects for carbon (Summons et al., 1998, Org. Geochem.). I believe this has implications for the inferences about the potential source of methane in the authors' study and should be discussed here.

Another point about the interpretation of the delta 13C data is that the authors appear to dismiss MPn as a precursor of methane because it is 'unlikely to be sufficiently 13C-depleted' as the 'organic matter' (is this DOM or POM?) in a nearby lake reached 'a minimum of -37‰'. I think this argument is not fully correct as (i) it is not clear whether MPn can be expected to be part of the measured 'organic matter' that had this isotopic signature, (ii) methane delta 13C values appeared to vary between the lakes and could reach around -43‰ (Figure 3), but, most importantly, (iii) the value is an average for the DOM or POM fraction – individual molecules in that pool can easily be much lighter than the measured value. Therefore, this argument needs to be reconsidered.

At last, the statement that MPn does not serve as methane precursor based on delta13C data appears to contrast with the statement in the abstract (line 28-29): 'these proteobacteria also metabolized methylphosphonate'. The authors should clarify what is the proposed role for MPn-derived methane production in this system.

Line 428ff: I don't understand the first sentence of this paragraph. What is meant with a 'methane source that occurs via an unknown pathway'?

How do the high rates in the BES treatment support the hypothesis of photosynthetic CH₄ production? BES is a methanogenesis inhibitor and I do not immediately see a direct or indirect link with photosynthesis.

Line 460-464: 'these findings suggest that two possible mechanisms for paradoxical CH₄ production co-occur.' and in the next sentence 'our data are indicative of an additional paradoxical CH₄ production mechanism ... by AAnP.' The authors should keep in mind that they discussed in their paper several different mechanisms and therefore need to be more concrete as to which 'two possible mechanisms' they mean here.

Figure 1: Are these rates based on the net CH₄ concentration difference between T_{zero} and T_{final}? Please clarify.

Figure 3/Line 897: 'In panel d, triangles denote UC3 BES treatment'. Panel d has no triangles in my version of the figure, instead panel c has a triangle at ca. 60 hours.

Reviewer #3 (Remarks to the Author):

This is a revised version of a manuscript I previously reviewed. The manuscript presents observations of methane production in oxic water column samples; these observations seem to be robust (though highly variable) and add to the evidence in the literature for oxic methane production. But the rest of the manuscript tries to make the case that particular bacteria and particular enzymes are responsible for the methane production; this material is unconvincing and speculative. Unfortunately, I do not think the revised manuscript adequately addresses previous concerns about overreaching interpretations and inconclusive results. The manuscript attempts to tie together many disparate observations that often do not logically fit together (esp some of the genomics results). While it is provocative, I do not think the conclusions are supported by the data.

specific concerns:

1. Title and manuscript suggest specific bacteria are implicated, but I find this evidence unconvincing. Comamonadaceae (including Limnohabitans) and Polynucleobacter are abundant in lakes. AAP are abundant in lakes. But the evidence does not clearly demonstrate methane production by any specific group.

2. I found the first section of the results confusing. Unclear which were 24 or 96 hour experiments, what "later" experiments were, redundancies in the text (e.g. 132-136 &

143-145, 138 & 148 "metatranscriptomes", 131-132 & 160-161)

3. Line 171-5 gives a mixed message – there is indeed some evidence that traditional methanogenesis is occurring but 173-5 seem to dismiss it. What does it mean that "these data provide limited evidence – but not the complete absence of evidence"? It seems the authors want to diminish it because it doesn't help their case for "paradoxical" production.

4. phnJ section: Line 192-3 doesn't make sense ("The majority of phnJ and phnD genes were present in organisms of the same genera as the metatranscriptomes"). The authors find phnJ in 40 bacterial genera, but "predominantly" in Comamonadaceae despite being only 20-30% of sequences. It seems the authors are trying to highlight Comamonadaceae but it is not clear the evidence actually supports this. I disagree with the conclusion (line 250-3) that "Our data in particular implicate multiple genera within the Comamonadaceae". These taxa are abundant in many lakes and are therefore abundant in sequence data.

5. Sequence identity, highlighted in several places in the manuscript (phnJ 200-203, 16S 224-226): I do not see the point of this information. Why is it relevant? What is the take-home message? High percent identity with reference sequences just indicates that there are good representatives in the database.

6. line 278-280: why would BES stimulate paradoxical methane production?

7. Line 317-320: are these the only 2 functions that are differentially expressed in the treatments vs control? Where are complete data?

8. Line 329-332: as with above – Limnohabitans and Polynucleobacter are abundant freshwater bacteria, so it is not surprising that they would contribute significantly to DPOR and COR sequences. But this alone does not implicate them in methane production.

9. Line 342-352: it makes sense that pigment genes might be differentially regulated in response to high light treatment. But again there is no demonstration that these genes are responsible for the correlated methane production. Correlation is not causation.

10. Line 369-371: this finding is not relevant or helpful. Yes, it is known that AAP and Limnohabitans are abundant in global lakes.

11. Line 372-387 is speculation that is not well supported by the data.

12. Line 462-464 "our data are indicative of" – no, the data do not clearly support methane production by AAP.

13. Metagenomics and metatranscriptomics methods are insufficient. I am not convinced the results are robust. Also it is not clear the utility of these data in the manuscript. Apart from expression of specific target genes, the rest of the genomics does not really add anything to the paper except confusion.

14. Figure 1: why are only select treatments shown? The legend says those without significant consumption or production are not shown, but yet there are some points shown on the zero line. What's the difference? Many of the treatments/experiments shown in Figure 2 are not plotted on Figure 1, so you cannot compare whether methane was produced with the expression data.

Perez-Coronel and Beman - Second Response to Reviewers

We thank the reviewers for their thoughtful comments on our revised manuscript. We have addressed all of their new comments in our second revision, which we think has further improved the manuscript. Our changes in the revised manuscript are highlighted in blue, and our specific responses to all reviewer comments are shown in **blue font** below.

REVIEWER COMMENTS

Reviewer #1 (Remarks to the Author):

I commend the authors for their detailed answers to the previous questions/ comments, and careful and extensive revision of the manuscript. I find the revised manuscript much improved from the previous version; key messages are now presented clearly and easy to follow; figures and tables are also improved.

Overall this manuscript presents some new and interesting insights into the "paradoxical" oxic methane production in freshwaters. Granted, there are still many unknowns, but this study is a welcomed addition to this field of research.

We thank the reviewer for these positive comments, and we hope that our manuscript does contribute new useful information to the field.

The authors have addressed most of my previous comments satisfactorily, and I have only a couple of rather minor suggestions--- they are indeed only minor suggestions; the authors may or may not take them into consideration.

line 44: Change "indicates" to "may indicate". There are, frankly, still many who reject the notion that CH₄ can be produced under oxic conditions.

This is a good point, and we made this change.

line 208: "We found that *aat* sequences from all but..." Do you mean all bacterial genera?

We apologize for the lack of specificity here. We are referring to 'all' of the bacterial groups reported in the recent study by Wang et al., but not all bacterial genera. We revised and clarified this to state that *aat* sequences were from "four of these five groups" and listed the groups.

line 296: "enrich the remaining CH₄ pool" Do you mean enrich the isotopic composition of the remaining CH₄ pool?

This is what we meant and apologize for not including this detail. We added "isotopically" before "enrich."

line 312-387 and related Discussion: The results related to DPOR and COR are indeed very interesting and they support the idea of a photosynthesis-driven oxic CH₄ production pathway. However, based on your explanation, DPOR and COR actions do not directly produce CH₄ (nor based on my own understanding of the functions of DPOR nad COR). Likewise, it is still unclear to me how to explain the positive effect of BES on CH₄ production in this case. Are you suggesting that BES acts as electron donor in the reaction? On line 380 it is suggested that adding BES causes "stress" but this seems rather vague. Overall, I think you have very intriguing and worthy observations but there may be still missing pieces in the puzzle. Perhaps add a few words to line 468-472 to point readers to some future research questions.

We thank the reviewer for these thoughtful comments. In our original submission, we raised the possibility that BES acts as a source of electrons, but we removed this idea due to the comments of the reviewers during the first round of review. Given the comment above, and that Reviewers 2 and 3 also inquired about BES below, we added a supplementary note to our revised manuscript. In the main text, we note that "Although BES and high light intensity produced surprisingly similar patterns in CH₄ production, $\delta^{13}\text{CH}_4$ values, and gene expression (Figs. 3-5), there are several possible explanations for why BES may induce a response" and then refer to the Supplementary Note 3. We placed this in the second to last paragraph of the Discussion, but it could be relocated in the manuscript.

In Supplementary Note 3 we outline several possibilities for why BES may lead to CH₄ production, which include the fact that BES is not totally specific and has documented effects on other organisms and processes (including multiple references to support this); that other sulfonates are also known to have additional effects; that sulfonate metabolism appears to be important for aquatic microbes in general; and that BES is an artificial substance added in high concentrations that may induce stress. We draw an analogy to what is known from plants, where multiple stressors lead to a common response. We think that this information is more appropriate in the SI, but we would be happy to include it in the main paper.

Finally, we followed the recommendation of the reviewer and added a sentence to the final paragraph that includes several future research questions.

Reviewer #2 (Remarks to the Author):

Review of Perez-Colonel and Beman (NCOMMS-21-09510A)

This is a revised version of a manuscript I reviewed earlier. I very much appreciate the effort the authors put into amending the manuscript according to mine and other reviewers' suggestions, and I believe these changes have now made the manuscript better focused and stronger.

1. The authors have addressed the concern about the 'novelty' of the methane-producing mechanisms by now replacing 'novel' with the term 'paradoxical', which I agree is better. They also adjusted the main claims in the abstract and throughout the text to better reflect their results and data.

2. The link between Comamonadaceae, their potential involvement in methane production, and their distribution in aquatic systems is spelled out more clearly now and easier to follow.
3. The reproducibility issue has been addressed by including standard errors into the graphs.
4. The distribution and prevalence of the genetic potential for MPn utilization (ie. the presence and expression of the phnJ gene) is described more clearly and is now discussed in the context of the ecological data.
5. My reservations about the 'novel' DPOR/COR-based mechanism have also been partially addressed. The authors toned down some of their more speculative conclusions, and now more carefully outline their evidence regarding the potential involvement of these enzymes during photosynthesis-linked methane production. It seems that we agree that the link ultimately tying these enzymes to methane production is missing but the authors present enough data to allow for this interesting speculation.

We thank the reviewer for these comments and are pleased that our previous revision addressed her/his previous comments.

My biggest outstanding comment is that the discussion of the multiple proposed pathways and the many lines of evidence for and against the various mechanisms remains rather convoluted. The authors seem to be aware of this problem and made some improvements during the revision; nonetheless, some conclusions still do not come across clearly, the biggest one for me perhaps – what is the main process/organisms contributing to aerobic methane production in this system? Is there even a main process or are all the different processes more or less equal? Based on the title 'methane production associated with photosynthesis and nutrient acquisition' I get the feeling it is rather the latter (possibly with the exception of methanogenesis) but more clarity regarding this would be needed.

We completely agree that we are presenting a range of experimental data and results, and that this becomes a challenge to interpret. However, we think it is important to consider the full complexity of paradoxical CH₄ production, as this is critical for understanding this phenomenon and its future implications. For better or for worse, this means examining and discussing all of the potential production mechanisms and pathways in a comprehensive way. As the reviewer notes, one of our overall points is that multiple mechanisms produce CH₄ in the lakes that we studied. Our other main point is that we find evidence for a mechanism associated with DPOR and COR. Obviously these two points are intertwined, and we think both are important.

In the previous version of the manuscript, we did note that MPn may represent a 'baseline' source of CH₄ (at the beginning of the second paragraph of the Discussion). We then discussed variability in photosynthesis at the end of the fourth paragraph of the Discussion. We noted that this may be superimposed on top of MPn and MeA-based production, and then mentioned CH₄ oxidation.

However, we agree that presenting all of this together in just a few sentences was confusing. We split up the sentence at the end of the fourth paragraph of the Discussion, and revised it and the text around it. We also added text before and after. We now note that photosynthetic

production was variable across our experiments and treatments. We then sum up by noting that we might expect periodic pulses of photosynthetic CH₄ production, consistent CH₄ production from nutrient acquisition, and rare production from methanogenesis. Finally, we note that research in additional lakes should determine whether this overall pattern holds elsewhere.

In addition to these changes in the Discussion, note that we revised the title to address the comments of Reviewer 3 below.

Other comments:

Abstract, Line 29: evidence is not provided that Proteobacteria 'metabolized methylphosphonate' but that they encoded and expressed the genes to degrade methylphosphonate.

We changed this text to directly follow the reviewer's suggestion.

Line 46: Please check. It was my understanding that the 'methane paradox' pertains to methane supersaturation in oxic water not its production per se - albeit it might be used interchangeably now.

This is a good point and we changed this to refer to "the accumulation of CH₄ under oxic conditions" rather than to its production.

Line 73: 'similar' is misleading here. What specifically is the similarity referring to? It is not the enzymatic machinery, neither the regulation.

We revised this sentence to remove "a similar mechanism may produce," and instead simply state the findings of the recent paper by Wang et al.

Line 108: CH₄ concentration measurements, presumably?

We apologize for the lack of detail here and added "concentration" to this sentence.

Line 120-121: sentence 'Given these consistencies...' is in my opinion redundant. It appears to suggest that due to the similarity in the magnitude of the rates, the rates appear broadly relevant, which does not make sense to me.

We removed this sentence, as we agree it is redundant with the information provided in the previous sentences.

Line 122-125: It is unclear what the 'initial' and 'later' experiments refer to? In the figures and throughout the text the experiments are described under their acronyms (eg. LG1, UC) so it is not immediately obvious which ones are considered 'initial' and which ones 'later'. Is this distinction even important? Is it not enough to say some incubations were 24 hours and some up to 96 hours long?

The reviewer makes a good point here. We removed “initial” and “later,” and now simply refer to 24 hour experiments vs. longer experiments. We also added a reference to the online table that lists the experiments and their lengths (as well as other aspects of the experiments).

Line 160: add ‘methane’ between ‘showed’ and ‘consumption’ to avoid ambiguity.

We thank the reviewer for the suggestion and added “CH₄” between “showed” and “consumption.”

Line 171: ‘Collectively, these data provide limited evidence...for water column-based methanogenesis as source of CH₄.’ This sentence needs additional clarification. What type of methanogenesis do you mean, ie. which carbon compound do you presume would be the precursor of CH₄? Archaeal methanogens can produce CH₄ from CO₂, acetate or methylated compounds such as methyl-amines and methyl-sulfides. Please elaborate.

We agree that it is important to distinguish between hydrogenotrophic, acetotrophic, and methylotrophic methanogenesis—particularly later in the paper—and so we included this information here. We added a sentence noting that 16S and *mcrA* sequence data are consistent with hydrogenotrophic and acetotrophic (but not methylotrophic) methanogenesis, and cite Grossart et al. (2011) and a recent and relevant paper by Li et al. (2021) in *Environmental Microbiology*.

(Note that in response to Reviewer 3 below, we also modified and split the sentence quoted here by Reviewer 2. We now state that there is some evidence for methanogenesis in the LG5 experiment, but not in others. Please see our response to Reviewer 3 below for additional specifics.)

Line 178-179: ‘other than conventional methanogenesis’ is misleading. Classical euryarchaeal methanogens are also capable of using methylamines as methane precursors (ie. methylotrophic methanogens). The authors presumably want to highlight that methylamine should be demethylated via an aerobic - non-methanogenic - pathway. This should be clarified.

We agree that the first sentence of this section was unclear following the subtitle. We made three changes, adding “Aerobic” to the subtitle, deleting “other than conventional methanogenesis” from the first sentence, and adding “multiple additional” before “CH₄ production mechanisms.”

Line 186: what is meant with ‘universally’?

Our meaning was that this gene was expressed in all metatranscriptomes. We now simply state this rather than use the word “universally.”

Line 188: remove 'and active'. 'Active' would traditionally refer to enzymes and proteins rather than transcripts.

We changed "active" to "expressed" in order to clarify this.

Line 215: '16S data' should read '16S rRNA gene data'.

Here we were trying to simplify the text, but we agree and have changed this to "16S rRNA gene and transcript data" to be more specific.

Line 253-255: Sentence 'In addition, ...' should be removed from here. The sentence describes the recovery of *aat* genes but is placed in a paragraph that deals with methylphosphonate metabolism; this is confusing. Additionally, it is repetitive with the information that was provided in more detail in the paragraph on line 204ff.

We removed this sentence following the reviewer's suggestion.

Line 300: This paragraph somewhat confuses me. The authors discuss the measured methane $\delta^{13}\text{C}$ values and conclude that the data 'suggest another paradoxical CH_4 production mechanism'. First of all, it is unclear to me which other production mechanism this refers to. Is it the DPOR/COR pathway? In cyanobacteria or AAnP? What would be the substrate for the DPOR/COR pathway, CO_2 ? Do the authors propose this pathway would result in fractionation as high as methanogenesis?

We apologize for the confusion here. Some of this is stylistic, as the very next sentence goes on to discuss CH_4 production by Cyanobacteria, which is the "...other paradoxical CH_4 production mechanism" that we mention at the end of this paragraph. It is also important to note that, at this point in the manuscript, we have not yet presented or discussed the DPOR/COR etc. results. Here we are presenting the ^{13}C data, using them to evaluate potential production mechanisms, and then setting the stage for the presentation of our omic results starting in the next paragraph. However, we revised the last sentence of this paragraph to be more specific, adding that "similar results have been interpreted as evidence for CH_4 production by photosynthetic organisms," and citing a very recent paper by Thottathil et al. in *Environmental Science and Technology*.

We also added a citation to this paper and modified a sentence in the Discussion that discusses the isotopic data. (Note that we discussed fractionation in our first response to the reviewers, and, to be brief, it does seem that fractionation could be comparable to, though not as high as, methanogenesis. CO_2 is one possible substrate that we do mention in the paper at the end of the Results; it is also possible that the CH_4 is derived from methoxyl groups, which we also mention in the paper.) We modified the sentence in the Discussion to directly state the possible explanations for the observed $\delta^{13}\text{C}_{\text{CH}_4}$ values, and then note that they are most consistent with the interpretation in the recent paper.

Next, related to the methanogenesis pathway, as mentioned earlier, the authors seem to use this term interchangeably with methanogenesis from CO₂. However, there are three 'classical' methanogenic pathways, which use CO₂, acetate and methylated compounds, respectively. These pathways result in different isotope effects for carbon (Summons et al., 1998, Org. Geochem.). I believe this has implications for the inferences about the potential source of methane in the authors' study and should be discussed here.

This is a good point. We now note that these values are consistent only with acetoclastic methanogenesis, and refer to a recent thorough review/book chapter by Whiticar, which is an update of the extensive Whiticar 1999 review.

Another point about the interpretation of the delta 13C data is that the authors appear to dismiss MPn as a precursor of methane because it is 'unlikely to be sufficiently 13C-depleted' as the 'organic matter' (is this DOM or POM?) in a nearby lake reached 'a minimum of -37‰'. I think this argument is not fully correct as (i) it is not clear whether MPn can be expected to be part of the measured 'organic matter' that had this isotopic signature, (ii) methane delta 13C values appeared to vary between the lakes and could reach around -43‰ (Figure 3), but, most importantly, (iii) the value is an average for the DOM or POM fraction – individual molecules in that pool can easily be much lighter than the measured value. Therefore, this argument needs to be reconsidered.

We agree in general with these comments and have revised this accordingly, but there are some important details to consider. First, regarding point (ii), we are focusing on the results showing production of lighter CH₄. It is true that higher $\delta^{13}\text{C}_{\text{CH}_4}$ values were measured in the LG6 and L7 experiments, but there was no significant CH₄ production in those experiments. Throughout this paragraph, we clarified that we are referring to the production of CH₄ of ca. -55 per mil.

Re point (i), we do agree that MPn may not be in the DOM or POM pool, as it could be rapidly metabolized. We deleted the specific mention of the $\delta^{13}\text{C}$ of OM in the nearby lake.

Re point (iii), we agree that different pools of OM can have different isotopic values, but typically these variations are not so extreme. -55 per mil is quite low, approximately 30 per mil lower than even terrestrial C3 plants. We have not seen any compound-specific isotope data to indicate that this could occur. However, we do agree that this is possible, just unlikely. We rephrased this part of the paragraph and added some additional information for clarification. First, we changed the phrasing to "In order for MPn metabolism to produce..." rather than "MPn is unlikely to produce..." We changed "is" to "seems" in the next sentence. We then note that the C atom in MPn is directly derived from PEP and refer to a detailed review of the biochemistry of phosphonates. We then note that PEP is actively cycled in the cell but would need to be depleted by 10s per mil in comparison with overall cellular C in order to yield sufficiently 'light' MPn. We feel that this is the most parsimonious interpretation of the data, as we leave open the possibility that MPn is coming from an exceptionally light source, but it does seem unlikely given our data and other recent work.

At last, the statement that MPn does not serve as methane precursor based on delta13C data appears to contrast with the statement in the abstract (line 28-29): 'these proteobacteria also metabolized methylphosphonate'. The authors should clarify what is the proposed role for MPn-derived methane production in this system.

Please see our comments above w/re to the Reviewer 2's overall comment about the importance of different mechanisms, as we made several clarifications throughout the paper. The changes made to this paragraph (outlined above) should also help clarify this. Finally, we clarified this within the abstract, adding "key" before "experimental treatments," and adding "In other experiments" to the start of the sentence that summarizes the MPn-related findings.

Line 428ff: I don't understand the first sentence of this paragraph. What is meant with a 'methane source that occurs via an unknown pathway'? How do the high rates in the BES treatment support the hypothesis of photosynthetic CH₄ production? BES is a methanogenesis inhibitor and I do not immediately see a direct or indirect link with photosynthesis.

We agree that the first sentence of this paragraph was confusing; the second clause in the sentence refers to photosynthetic CH₄ production, which is the "newly-identified source that occurs via an unknown pathway." We modified the first portion of this sentence to directly state that photosynthesis represents a new but poorly understood source, and then we converted the second part of the sentence into a new sentence.

Re BES, please see our response above to Reviewer 1's final comment; in short, we added a sentence and corresponding Supplementary Note 3 to address this.

Line 460-464: 'these findings suggest that two possible mechanisms for paradoxical CH₄ production co-occur.' and in the next sentence 'our data are indicative of an additional paradoxical CH₄ production mechanism ... by AAnP.' The authors should keep in mind that they discussed in their paper several different mechanisms and therefore need to be more concrete as to which 'two possible mechanisms' they mean here.

We apologize for the lack of detail here. We now note that the two possible mechanisms that co-occur are "photosynthetic" and "phosphonate-based."

Figure 1: Are these rates based on the net CH₄ concentration difference between T_{zero} and T_{final}? Please clarify.

We agree that this wasn't clear; the rates are described in the methods but not in the figure caption, although the standard errors from the correlations are described in the figure caption. We added a sentence to the figure caption noting that the rates are from the slopes of significant correlations between CH₄ over time in the incubations.

Figure 3/Line 897: 'In panel d, triangles denote UC3 BES treatment'. Panel d has no triangles in my version of the figure, instead panel c has a triangle at ca. 60 hours.

We apologize for this typo and fixed this.

Reviewer #3 (Remarks to the Author):

This is a revised version of a manuscript I previously reviewed. The manuscript presents observations of methane production in oxic water column samples; these observations seem to be robust (though highly variable) and add to the evidence in the literature for oxic methane production. But the rest of the manuscript tries to make the case that particular bacteria and particular enzymes are responsible for the methane production; this material is unconvincing and speculative. Unfortunately, I do not think the revised manuscript adequately addresses previous concerns about overreaching interpretations and inconclusive results. The manuscript attempts to tie together many disparate observations that often do not logically fit together (esp some of the genomics results). While it is provocative, I do not think the conclusions are supported by the data.

specific concerns:

1. Title and manuscript suggest specific bacteria are implicated, but I find this evidence unconvincing. Comamonadaceae (including Limnohabitans) and Polynucleobacter are abundant in lakes. AAP are abundant in lakes. But the evidence does not clearly demonstrate methane production by any specific group.

We modified the title in response to this comment, and no longer refer to 'broadly-distributed' bacteria.

2. I found the first section of the results confusing. Unclear which were 24 or 96 hour experiments, what "later" experiments were, redundancies in the text (e.g. 132-136 & 143-145, 138 & 148 "metatranscriptomes", 131-132 & 160-161)

We apologize for the confusion here. Please see our response to Reviewer 2 above about the "initial" vs. "later" experiments. In brief, we now just refer to the lengths of the experiments, and reference the online table with this information.

We also agree that there are some redundancies in the text. Re the redundancies between previous lines 132-136 & 143-145, we think it important to make the point that methane oxidation can obscure production, as this is relevant for interpreting our data, and data from other studies (particularly because some researchers still consider oxic CH₄ production to be controversial). As we note, this is also where experimental treatments and molecular data of various kinds can provide insight. We revised and simplified the sentence on previous lines 143-145 to make it less redundant.

Re previous lines 138 and 148, we deleted “in metatranscriptomes” in the first sentence and revised the second sentence to remove the list of analyses, and now simply refer to “analyses.” We left previous lines 131-132 & 160-161 unchanged, as the later text was specifically requested in the first round of review to summarize these particular experiments.

3. Line 171-5 gives a mixed message – there is indeed some evidence that traditional methanogenesis is occurring but 173-5 seem to dismiss it. What does it mean that “these data provide limited evidence – but not the complete absence of evidence”? It seems the authors want to diminish it because it doesn’t help their case for “paradoxical” production.

We agree that this sentence sent a mixed message, but we weren’t intending to diminish water column-based methanogenesis. As Grossart et al. originally demonstrated, this is still a form of paradoxical CH₄ production. While it isn’t a ‘new’ pathway, it is another way that CH₄ may be produced in the oxygenated water column and is very interesting.

Instead, our aim with this sentence was simply to avoid over-interpreting the data in either direction. We agree that the previous sentence was too vague, and so we modified this sentence and split it into two sentences to briefly summarize the evidence for and against. We note that there is limited evidence for water column-based methanogenesis from the LG5 incubation, where BES had significant effects and both 16S and *mcrA* were detected from methanogens. We then note that the other incubations showed CH₄ production but methanogens were almost entirely absent and inactive.

4. *phnJ* section: Line 192-3 doesn’t make sense (“The majority of *phnJ* and *phnD* genes were present in organisms of the same genera as the metatranscriptomes”). The authors find *phnJ* in 40 bacterial genera, but “predominantly” in Comamonadaceae despite being only 20-30% of sequences. It seems the authors are trying to highlight Comamonadaceae but it is not clear the evidence actually supports this. I disagree with the conclusion (line 250-3) that “Our data in particular implicate multiple genera within the Comamonadaceae”. These taxa are abundant in many lakes and are therefore abundant in sequence data.

We apologize for any confusion here, as the first sentence of this paragraph was a holdover from earlier versions of the manuscript. We deleted this first sentence, since it was not critically important.

We made additional clarifications in this section. We removed “predominantly” and replaced this with “most commonly.” We agree that 20-30% doesn’t seem predominant, but because there are so many genera, Comamonadaceae *phnJ* transcripts and genes were most common overall. They were also most common in all samples except those with high numbers of Sphingobacteriales, so we now state this in the paragraph.

Finally, we modified the sentence at the end of the MPn/MeA section in several ways. Our point here was that, to our knowledge, *phnJ* has not been reported from aquatic Comamonadaceae in the context of oxic CH₄ production. As a result, the fact that these organisms are abundant in

many lakes and contain *phnJ* is actually an important point to make, as it suggests the capability to metabolize MPn is widespread in lakes. While this was our original intention, we agree that our wording and use of “implicate” wasn’t effective. We made the following changes:

- We switched the word order at the very beginning of the sentence
- We deleted “implicate” and replaced this with “suggest that”
- We added information from the literature on the *Comamonadaceae*
- We modified the sentence to now directly state that the *Comamonadaceae* may utilize MPn
- We noted that high identities are “to sequences collected in other locations”
- And we also deleted the last portion of the sentence.

5. Sequence identity, highlighted in several places in the manuscript (phnJ 200-203, 16S 224-226): I do not see the point of this information. Why is it relevant? What is the take-home message? High percent identity with reference sequences just indicates that there are good representatives in the database.

We made several clarifications to address this comment. The key point isn’t that similar sequences are in the databases, but that these database sequences were collected in other real-world locations, indicating that similar genes/organisms are found in other locations. In addition to the change we made directly above noting high identities to sequences collected in other locations, we added this information to the second paragraph of the Discussion as well.

6. line 278-280: why would BES stimulate paradoxical methane production?

Please see our response above to Reviewer 1’s final comment; we added a sentence and corresponding Supplementary Note 3 to address this.

7. Line 317-320: are these the only 2 functions that are differentially expressed in the treatments vs control? Where are complete data?

We added a Supplementary Note 2 describing the few other functions that were significantly different. In short, two other functions were different in both BES and high light. One of these is also involved in chlorophyll metabolism (magnesium chelatase), while the other function may also be involved (geranylgeranyl diphosphate reductase). Another function (beta glucosidase) was also significantly different in the BES treatment, but not in the high light treatment, while several functions were higher under high light but not in the BES treatment.

8. Line 329-332: as with above – Limnohabitans and Polynucleobacter are abundant freshwater bacteria, so it is not surprising that they would contribute significantly to DPOR and COR sequences. But this alone does not implicate them in methane production.

To be clear, this sentence reports the relative abundances of DPOR and COR, and then notes that Limnohabitans and Polynucleobacter are AAnP. We do not refer to methane production here. We modified this sentence to note that “...DPOR and COR were most commonly found within...” rather than “...particularly common within...”

9. Line 342-352: it makes sense that pigment genes might be differentially regulated in response to high light treatment. But again there is no demonstration that these genes are responsible for the correlated methane production. Correlation is not causation.

We fully agree that correlation is not causation, but obviously it can still provide important insights. This is also an oversimplification of what we are presenting, as we discuss our results in the context of the literature. The fact is that CH₄ is produced and the treatment effects are large. It is unlikely that CH₄ is coming from methanogenesis or MPn, and so this requires explanation. Even in the absence of any other knowledge, correlation or other approaches may provide insight by determining which genes differ between treatments. This general approach is obviously widely used with RNAseq data etc.

However, there is substantial additional context to our results because photosynthetic organisms have been shown to produce CH₄, yet the pathway remains unknown. We observed a strong response for particular genes involved in photosynthesis. If we didn't already know that photosynthesis can produce CH₄, we might interpret these results differently. But this is known, and so the most logical explanation is that DPOR and COR are involved in photosynthetic CH₄ production. Based on the literature and our new results, any alternative explanation is far less logical.

There are other important points to consider, from what is known about CH₄ production in plants, to the fact that DPOR and COR are nitrogenase-like enzymes. There is recent, interesting, and relevant literature on both of these topics, as CH₄ production (and eventual flux to the atmosphere) is such an important issue. Correlation is not causation, but our results fit into the broader context of the literature. Most importantly, any other explanation requires discounting or ignoring all of this, and then invoking some other as-yet unidentified mechanism or pathway. If the reviewer is aware of any additional mechanisms that we should examine, we are happy to do so.

10. Line 369-371: this finding is not relevant or helpful. Yes, it is known that AAP and Limnhabitans are abundant in global lakes.

We added this sentence/brief analysis in response to previous reviewer comments. In response to this new comment, we deleted this sentence, deleted the corresponding sentence in the methods, and deleted the corresponding online table. We also moved the sentence prior to this, placing it earlier in this section.

11. Line 372-387 is speculation that is not well supported by the data.

Here we are interpreting our results in light of what is known from the literature. We present several possibilities based on earlier published findings. Again, we are happy to consider other alternative explanations, preferably based on published findings.

12. Line 462-464 “our data are indicative of” – no, the data do not clearly support methane production by AAP.

Obviously we respectfully disagree with the reviewer here, and please see several of our comments above.

13. Metagenomics and metatranscriptomics methods are insufficient. I am not convinced the results are robust. Also it is not clear the utility of these data in the manuscript. Apart from expression of specific target genes, the rest of the genomics does not really add anything to the paper except confusion.

We disagree with this, as we used several well-known software programs to analyze the data in multiple ways. We used a gene-centric approach, we assembled metagenomes, we used IMG to provide a broad analysis of key samples, and we mapped reads to assemblies and MAGs. As we mentioned in the previous round of review, we are more than happy to provide more information or conduct additional analyses. If the reviewer has concrete suggestions for improvement, we would be delighted to implement them.

We also disagree about the utility of the omics data. As we highlight in the paper and outline above, these data are important for providing insight into CH₄ production mechanisms. Without this information, it is completely unclear how CH₄ is produced. Even studies simply showing CH₄ production are, of course, very interesting and relevant given the importance of this topic. However, we provide an additional layer of insight into how CH₄ is produced, and this is provided primarily from omics.

14. Figure 1: why are only select treatments shown? The legend says those without significant consumption or production are not shown, but yet there are some points shown on the zero line. What's the difference? Many of the treatments/experiments shown in Figure 2 are not plotted on Figure 1, so you cannot compare whether methane was produced with the expression data.

We agree that this was unclear on the figure. Some treatments had low but significant rates, and so are shown on Figure 1. Others were statistically insignificant and were not shown. Without any markings, we agree that it is difficult to visually match up the different experiments and treatments. To clarify which treatments were not significant, we added asterisks to the relevant locations on the figure.

Reviewer #2 (Remarks to the Author):

Review of Perez-Colonel and Beman (NCOMMS-21-09510B): Multiple sources of paradoxical methane production in aquatic ecosystems include bacterial photosynthesis

Overall, I am satisfied with the changes the authors did to the manuscript during the second round of revision and how they addressed the comments I raised in my previous review.

I think this study is a valuable contribution to the current research on methane production in oxic surface waters. The authors' speculation about an involvement of a novel pathway in aerobic methane production presents an intriguing new hypothesis that should be further explored.

I only have a few minor comments:

Line 29-31: Regarding the sentence: 'In other experiments, Proteobacteria encoded and expressed genes to degrade methylphosphonate, while both photosynthetic- and methylphosphonate-based CH₄ production pathways occurred together within metagenome-assembled genomes.'

- It is not clear what 'in other experiments' refers to, leave out or specify.

- The formulation 'pathways occurred ... within genomes' is imprecise. Maybe better '... while genes encoding for both pathways co-occurred in metagenome-assembled genomes?'

Line 92: 'We developed an experimental approach to disentangle these mechanisms'

- The authors' approach of combining experimental, isotopic and molecular work is surely commendable but not novel; therefore, it should not be presented as something they newly developed. Better to replace with 'we applied an interdisciplinary approach'.

Line 189: 'Finally, we examined whether transcripts and genes involved in the production of MPn were also present and expressed.'

- Removing 'transcripts' will make the sentence correct.

Line 207: 'Additional isolates with this ability included Pseudomonas, Caulobacter, Mesorhizobium, and Dietzia spp.—but this paradoxical CH₄ production mechanism has not been examined elsewhere.'

- it's unclear what 'elsewhere' refers to in this context, is it these bacterial groups or different geographic location? Please remove or clarify.

Line 217: '16S rRNA gene and transcript data were consistent with metatranscriptomic and metagenomic data.'

- I suppose the authors mean to compare the 16S rRNA gene and transcript data to functional gene analyses, rather than metatranscriptomic and metagenomics data in general?

Line 279: 'This counterintuitive effect suggests another source of CH₄.'

- The meaning of this sentence is unclear to me.

Line 321: 'This is an advantage of -omic data, as it is possible to examine previously unidentified potential mechanisms for CH₄ production.'

- I disagree with this statement: -omics analyses rely on cross-reference with genes of known (or at least assigned) function. Hence, I can't see how such approach can discover something novel. However, I can see how in this study the -omics data allowed to explore processes for which no net activity was measured, such as methane oxidation. As far as I'm concerned, that was in this case the true advantage of -omic data.

Reviewer #3 (Remarks to the Author):

The authors have addressed most of my major concerns. We may not agree on all aspects of the data interpretation, but I do believe this is an intriguing manuscript that presents new ideas and thereby advances the field. I appreciate the new supplementary notes that discuss, in more detail, possible explanations.

Perez-Coronel and Beman - Final Response to Reviewer(s)

We thank the reviewers for their thoughtful comments on our revised manuscript. We have addressed all of Reviewer 2's additional comments in our final revision, which we think has further improved the manuscript. Our changes in the revised manuscript are highlighted in blue, and our specific responses to all reviewer comments are shown in blue font below.

REVIEWER COMMENTS

Reviewer #2 (Remarks to the Author):

Review of Perez-Colonel and Beman (NCOMMS-21-09510B): Multiple sources of paradoxical methane production in aquatic ecosystems include bacterial photosynthesis

Overall, I am satisfied with the changes the authors did to the manuscript during the second round of revision and how they addressed the comments I raised in my previous review.

I think this study is a valuable contribution to the current research on methane production in oxic surface waters. The authors' speculation about an involvement of a novel pathway in aerobic methane production presents an intriguing new hypothesis that should be further explored.

I only have a few minor comments:

Line 29-31: Regarding the sentence: 'In other experiments, Proteobacteria encoded and expressed genes to degrade methylphosphonate, while both photosynthetic- and methylphosphonate-based CH₄ production pathways occurred together within metagenome-assembled genomes.'

- It is not clear what 'in other experiments' refers to, leave out or specify.
- The formulation 'pathways occurred ... within genomes' is imprecise. Maybe better '... while genes encoding for both pathways co-occurred in metagenome-assembled genomes?'

We closely followed the recommendation of the journal editors in revising our abstract, and the first comment no longer applies to this sentence. The revisions closely follow the second comment to state that genes encoding for both pathways co-occur in MAGs.

Line 92: 'We developed an experimental approach to disentangle these mechanisms'

- The authors' approach of combining experimental, isotopic and molecular work is surely commendable but not novel; therefore, it should not be presented as something they newly developed. Better to replace with 'we applied an interdisciplinary approach'.

We changed this to state that "Here we use an interdisciplinary approach..."

Line 189: 'Finally, we examined whether transcripts and genes involved in the production of MPn were also present and expressed.'

- Removing 'transcripts' will make the sentence correct.

We removed 'transcripts.'

Line 207: 'Additional isolates with this ability included Pseudomonas, Caulobacter, Mesorhizobium, and Dietzia spp.—but this paradoxical CH₄ production mechanism has not

been examined elsewhere.'

- it's unclear what 'elsewhere' refers to in this context, is it these bacterial groups or different geographic location? Please remove or clarify.

We revised this to specify that this "has only been examined in a single location" rather than "has not been examined elsewhere."

Line 217: '16S rRNA gene and transcript data were consistent with metatranscriptomic and metagenomic data.'

- I suppose the authors mean to compare the 16S rRNA gene and transcript data to functional gene analyses, rather than metatranscriptomic and metagenomics data in general?

We revised this sentence to state that "16S rRNA gene and transcript data were consistent with functional gene data from metatranscriptomes and metagenomes."

Line 279: 'This counterintuitive effect suggests another source of CH₄.'

- The meaning of this sentence is unclear to me.

We now clearly spell out our meaning for those that cannot easily infer it; the sentence now reads "This counterintuitive effect of increasing CH₄ production—despite the addition of a known methanogenesis inhibitor—suggests another source of CH₄."

Line 321: 'This is an advantage of 'omic data, as it is possible to examine previously unidentified potential mechanisms for CH₄ production.'

- I disagree with this statement:—omics analyses rely on cross-reference with genes of known (or at least assigned) function. Hence, I can't see how such approach can discover something novel. However, I can see how in this study the —omics data allowed to explore processes for which no net activity was measured, such as methane oxidation. As far as I'm concerned, that was in this case the true advantage of —omic data.

We removed this sentence.